# Integrative multi-omics profiling reveals cAMP-independent mechanisms regulating hyphal morphogenesis in *Candida albicans*

**Kyunghun Min**[1], **Thomas F. Jannace**[1¤a], **Haoyu Si**[1¤b], **Krishna R. Veeramah**[2], **John D. Haley**[3,4], **James B. Konopka**[1]*

**1** Department of Microbiology and Immunology, Renaissance School of Medicine, Stony Brook University (SUNY), Stony Brook, New York, United States of America, **2** Department of Ecology and Evolution, Stony Brook University (SUNY), Stony Brook, New York, United States of America, **3** Department of Pathology, Renaissance School of Medicine, Stony Brook University (SUNY), Stony Brook, New York, United States of America, **4** Biological Mass Spectrometry Shared Resource, Renaissance School of Medicine, Stony Brook University (SUNY), Stony Brook, New York, United States of America

¤a Current address: Department of Quality Assurance, Biogen, Morrisville, North Carolina, United States of America
¤b Current address: School of Biological Sciences, University of Nebraska-Lincoln, Lincoln, Nebraska, United States of America
* james.konopka@stonybrook.edu

**Data Availability Statement:** The datasets generated in this study can be found under NCBI BioProject PRJNA706677, including whole-genome sequence data and RNA-seq data. Source

## Abstract

Microbial pathogens grow in a wide range of different morphologies that provide distinct advantages for virulence. In the fungal pathogen *Candida albicans*, adenylyl cyclase (Cyr1) is thought to be a master regulator of the switch to invasive hyphal morphogenesis and biofilm formation. However, faster growing *cyr1Δ/Δ* pseudorevertant (PR) mutants were identified that form hyphae in the absence of cAMP. Isolation of additional PR mutants revealed that their improved growth was due to loss of one copy of *BCY1*, the negative regulatory subunit of protein kinase A (PKA) from the left arm of chromosome 2. Furthermore, hyphal morphogenesis was improved in some of PR mutants by multigenic haploinsufficiency resulting from loss of large regions of the left arm of chromosome 2, including global transcriptional regulators. Interestingly, hyphal-associated genes were also induced in a manner that was independent of cAMP. This indicates that basal protein kinase A activity is an important prerequisite to induce hyphae, but activation of adenylyl cyclase is not needed. Instead, phosphoproteomic analysis indicated that the Cdc28 cyclin-dependent kinase and the casein kinase 1 family member Yck2 play key roles in promoting polarized growth. In addition, integrating transcriptomic and proteomic data reveals hyphal stimuli induce increased production of key transcription factors that contribute to polarized morphogenesis.

## Author summary

The human fungal pathogen *Candida albicans* switches between budding and filamentous hyphal morphologies to gain advantages for virulence and survival in the host. Although

data are provided within the paper and its Supporting Information files.

**Funding:** This work was supported by Public Health Service grants from the National Institutes of Health (https://www.nih.gov) awarded to J.B.K. (R01GM116048 and R01AI047837). The funders had no role in study design, data collection and analysis, decision to publish, or preparation of the manuscript.

**Competing interests:** The authors have declared that no competing interests exist.

adenylyl cyclase has been thought to be a master regulator that controls this switch, we identified *C. albicans* pseudorevertant mutants that grow better and form hyphae in the absence of adenylyl cyclase and cAMP. The mutant cells were also able to induce hyphal-associated genes in the absence of cAMP that are needed for virulence. Integrating information from different omics approaches identified cAMP-independent mechanisms that promote hyphal growth. This includes phosphoproteomic studies that revealed key roles for the Cdc28 cyclin-dependent kinase and casein kinase 1 in promoting hyphal growth. In addition, integrating transcriptomic and proteomic data revealed that post-transcriptional mechanisms regulate the levels of a set of key transcription factors that are important for hyphal induction, suggesting a special type of translational regulation. These studies better define the pathways that stimulate *C. albican*s to switch from budding to hyphal growth, which is important for invasion into tissues, escape from the immune system, and biofilm formation.

## Introduction

Fungal pathogens are capable of transitioning between different morphologies that provide distinct advantages for virulence and survival in the host [1,2]. For example, the fungal pathogen *Candida albicans* switches from budding to hyphal growth, which promotes virulence as the long hyphal filaments mediate invasion into tissues, escape from immune cells, and biofilm formation [1–6]. A variety of stimuli promote hyphal morphogenesis, including serum, $CO_2$, alkaline pH, peptidoglycan breakdown products, N-acetylglucosamine (GlcNAc), and contact with a solid matrix [3,7,8]. In *C. albicans*, adenylyl cyclase (Cyr1) is thought to be a master regulator of hyphal growth [3,5,9–11]. Deletion of adenylyl cyclase (*cyr1Δ/Δ*) blocks hyphal formation, and addition of millimolar levels of exogenous cAMP induces it [12]. However, interpreting the role of Cyr1 and cAMP is complicated by the fact that the *cyr1Δ/Δ* mutants grow very poorly [13]. Furthermore, a recent study showed that faster growing *cyr1Δ/Δ* pseudorevertant (PR) mutants form hyphae in the absence of Cyr1 and cAMP [14]. Thus, although cAMP is capable of inducing hyphae, other pathways can contribute to stimulating the formation of filamentous hyphal cells [14].

In *C. albicans*, cAMP formed from ATP by Cyr1 acts by binding to the negative regulatory subunit (Bcy1) of protein kinase A (PKA), thereby releasing the PKA catalytic subunits (Tpk1 and Tpk2) to phosphorylate target proteins [11]. Most current models propose that the cAMP-PKA pathway induces the expression of a special set of genes that promote hyphal growth [2,3,11,13,15,16]. This has been supported by the ability of many transcription factors (TFs) to regulate this switch. For example, PKA was reported to phosphorylate the Efg1 TF, which is required for hyphal induction [17,18]. However, it has been surprising that none of the hyphal-induced genes has been shown to play a direct role in the morphogenesis machinery that determines cell shape [19,20]. The common core set of genes stimulated by a group of different hyphal inducers in *C. albicans* do not account for the transition to hyphal growth [19]. In addition, our recent studies showed that a mutant that cannot catabolize GlcNAc can be stimulated by GlcNAc to form hyphae without induction of hyphal-specific genes [20]. Other studies have shown that some hyphal regulatory TFs can be bypassed under special environmental conditions [21,22], or by altered expression of the Cak1 protein kinase [23], the absence of a variant histone H3 [24], or mutation of the Ssn3 subunit of the Mediator transcription complex [25]. This indicates that although TFs are important for creating the proper

physiological environment for cells to undergo hyphal growth, the target genes are not directly involved in promoting filamentous morphogenesis.

Other studies have implicated several protein kinases in hyphal signaling that are not directly regulated by cAMP [2,3,26]. For example, the cyclin-dependent kinase Cdc28 promotes hyphal growth in part by phosphorylating several proteins involved in polarized morphogenesis [26–34]. Therefore, to better define the cAMP-dependent and independent mechanisms of hyphal signaling, we isolated a set of new *cyr1*Δ/Δ PR mutants that displayed variable levels of improved growth rates and ability to form hyphae. The wild type and PR mutant cells were compared with four omics approaches: genomics combined with gene-mapping by CRISPR-Cas9 to identify genetic changes in PR mutants, transcriptomics to better define the role of cAMP in hyphal gene regulation, proteomics and phosphoproteomics to reveal key kinases for hyphal induction. Furthermore, integrative transcriptomics and proteomics profiling indicates that there are increases in the levels of key transcription factors, possibly due to changes in their translation rate, that are induced by hyphal signaling. Altogether, these results reveal how multiple processes in the gene expression pathway can modulate cellular differentiation.

## Results

### Large deletions in the left arm of chromosome 2 bypass the need for cAMP to improve the growth rate and to stimulate hyphal morphogenesis

We previously reported that slow growing *cyr1*Δ/Δ mutant strains give rise to faster growing cells, which we termed pseudorevertants (PRs) [14]. Interestingly, these spontaneous PRs could form hyphae even though they lack Cyr1 and cAMP [14]. To better define the underlying mechanisms, we isolated additional PR strains. These PR mutants were isolated from independent cultures which were started from different single colonies of the *cyr1*Δ/Δ strain. We found that they differed in their growth rates and ability to be induced to form hyphae (Figs 1A, 1B and S1). Based on these phenotypic differences, the new PR mutants could be grouped into 4 classes (Table 1). Surprisingly, the degree of improved growth rate of the PR mutants did not correlate with their ability to form filamentous cells, indicating that the mechanisms of suppression were complex. The faster growing Class 1 mutant doubled at a nearly wild-type rate (1.5 h) but only 74% of the cells formed filamentous cells. In contrast, the Class 2, Class 3, and Class 3+ PR mutants grew slower than the Class 1 mutants but were more efficient at forming filamentous cells. For example, Class 2 mutants grew at a doubling time of 1.8 h yet 93% of the cells were filamentous. Class 3 and 3+ produced filamentous cells but they were shorter and thicker than the WT hyphae.

Whole genome sequencing of the diploid *C. albicans* strains identified distinct genetic changes in the four classes of PR mutants. Interestingly, although they were different, all four classes of mutants contained genetic changes that resulted in loss of expression of one copy of *BCY1*, which encodes the negative regulatory subunit of cAMP-dependent PKA kinase. This suggested that reduced expression of *BCY1* increased PKA activity resulting in improved growth in the absence of cAMP. Similar results reported that a *bcy1*Δ mutation can rescue growth of a *cyr1*Δ mutant in *S. cerevisiae* [35]. Class 1 mutants exhibited the simplest genetic change, in that the improved growth correlated with a premature stop codon in *BCY1* on chromosome 2 (Fig 1C and S1 Table and S1 Data). In contrast, the Class 2, 3, and 3+ PR mutants all lacked part of the left arm of one copy of chromosome 2 that includes *BCY1*. We previously did not report a change in ploidy for PR2 [14]. However, higher read coverage of the genome sequences in the present study (>90×) aided in detecting changes in ploidy of chromosome 2 and in identifying the junction regions (Fig 1C and S1 Table). Inspection of the genome

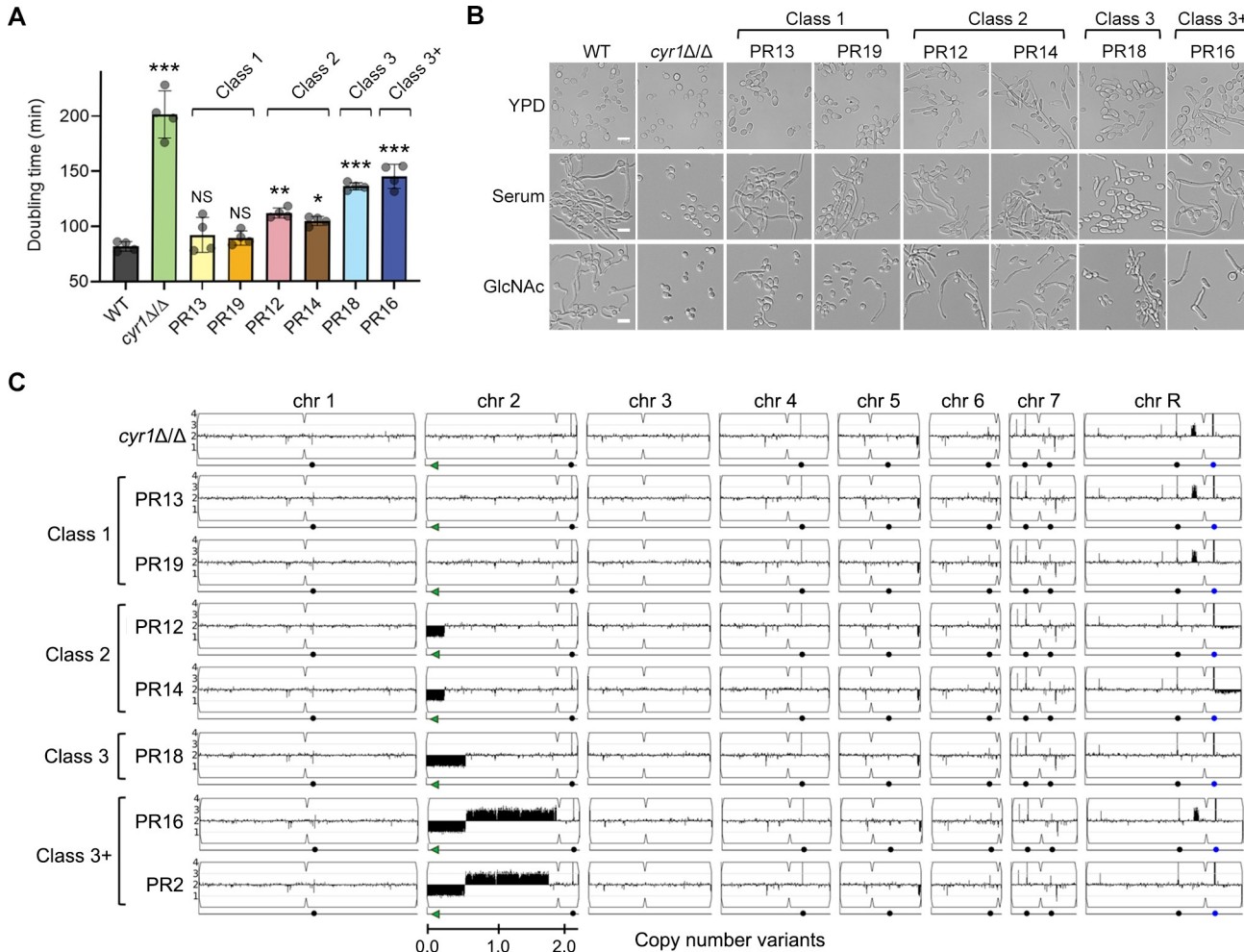

**Fig 1. Genetic changes in chromosome 2 bypassed the need for cAMP to improve growth and hyphal induction in *cyr1Δ/Δ* PR mutants.** (A) Doubling times were measured in liquid YPD medium at 30°C. Shown is the mean ± SD (standard deviation) of 4 independent experiments. Statistical analysis was performed using one-way ANOVA with Dunnett's multiple comparisons test comparing the strains with the wild-type control (WT); [NS] $p > 0.05$, [*] $p < 0.05$, [**] $p < 0.01$, [***] $p < 0.001$. The doubling time of the strains shown was significantly shorter when compared to the *cyr1Δ/Δ* background ($p < 0.0001$). (B) The strains indicated at the top were grown in the liquid medium indicated on the left, and then hyphal induction was assessed microscopically. Cells were grown in liquid medium containing 15% serum or 50 mM N-acetylglucosamine (GlcNAc) to induce hyphal growth. Cells were incubated at 37°C for 3 h and then photographed. Scale bar, 10 μm. (C) Copy number variation analysis based upon read depth across the genome. Copy number estimates scaled to genome ploidy (Y-axis) and chromosome location (X-axis) were plotted using YMAP [91]. Numbers and symbols below chromosomes indicate chromosomal position (Mb), *BCY1* gene (green arrows), centromere locus (indentations in the chromosome cartoon), major repeat sequence position (black circles), and rDNA locus (blue circles, ChrR).

sequences previously reported for PR3 and PR4 also indicates that they have altered ploidy for regions of chromosome 2 similar to the class 3+ PR mutants.

Western blot analysis showed that the Bcy1 protein levels were reduced by 50% in the PR mutants, confirming that altered gene dosage leads to reduced production of Bcy1 (S2 Fig). The reduction in Bcy1 did not impact the levels of Tpk2, one of the PKA catalytic subunits in the PR mutants compared to the WT. Interestingly, *TPK2* was located in the middle of the chromosome 2, which was duplicated in Class 3+ PRs. However, none of the other PKA pathway components were located on chromosome 2. This supports the conclusion that the haploinsufficiency of *BCY1* bypassed the need for cAMP for growth and hyphal induction in *C. albicans*.

**Table 1. Phenotypic and genetic variations of *cyr1Δ/Δ* pseudorevertants (PRs).**

| Strain | PR class | Doubling time (h)[a] | Filamentation (%)[b] | Short genotype[c] |
|---|---|---|---|---|
| WT | | 1.4 ± 0.07 | 98 ± 2.3 | Prototrophic wild type strain |
| *cyr1Δ/Δ* | | 3.4 ± 0.36 *** | 0.0 *** | *cyr1Δ/Δ* (parental strain) |
| PR13, PR19 | Class 1 | 1.5 ± 0.19 | 74 ± 4.4 *** | *bcy1*/*BCY1 cyr1Δ/Δ* |
| PR12, PR14 | Class 2 | 1.8 ± 0.09 ** | 93 ± 2.3 | Monosomy of ~276 kb of Chr2L; *cyr1Δ/Δ* |
| PR18 | Class 3 | 2.3 ± 0.06 *** | 93 ± 4.2 | Monosomy of ~590 kb of Chr2L; *cyr1Δ/Δ* |
| PR2, PR16 | Class 3+ | 2.4 ± 0.18 *** | 82 ± 16.4 ** | Monosomy of ~557 kb of Chr2L; trisomy of ~1.3 Mb of Chr2; *cyr1Δ/Δ* |

[a] Doubling times were measured in liquid YPD medium at 30°C. Shown is the mean ± SD of 4 independent experiments.

[b] The percent of filamentous cells was measured 3 h after growth in GlcNAc medium at 37°C. Shown is the mean ± SD of at least 3 independent experiments with at least 100 cells counted for each condition.

[c] An asterisk indicates a nonsense mutation. Chr2L, left arm of chromosome 2.

The p-value was calculated using one-way ANOVA with Dunnett's multiple comparisons test comparing the strains with the WT control

**p < 0.01

***p < 0.001.

In the Class 2 and 3+ PRs, similar breakpoints were identified in the independently isolated strains indicating that a specific DNA sequence induced the chromosomal deletion. For example, PR12 and PR14 (class 2) lost the same 276-kb region of chromosome 2 although they were isolated from independent cultures on different days. We hypothesized that a specific DNA sequence induced the breakpoints at the same location. To better understand how the large deletions of 270 kb and 590 kb were generated in the Class 2, 3, and 3+ PRs, we inspected the genome sequences for matches to the *C. albicans* 23-bp telomere repeat sequence (CACCAA-GAAGTTAGACATCCGTA) [36], which was found at the deletion end points (S1 Table). Analysis of the breakpoint regions in chromosome 2 revealed 4 to 9-bp matches to telomere seed sequences. However, the long repeat sequence, which was previously reported to drive genome rearrangements, was not found near the deletion sites [37]. As described previously, this suggests that chromosome break healing by *de novo* telomere addition contributes to the genome plasticity in *C. albicans* [38–40], which enables it to adapt to adverse conditions or deleterious mutations such as *cyr1Δ/Δ*.

## Gene mapping by CRISPR-Cas9 identified general transcription factors are involved in hyphal regulation

To confirm that the genetic changes in chromosome 2 improved growth and hyphal induction in the PR mutants, transient expression of the CRISPR-Cas9 system [41,42] was utilized to recreate the loss of one copy of *BCY1* and the large deletions present in the Class 2, 3, and 3+ PR mutants. To facilitate large deletions, two single guide RNAs (sgRNAs) were used to target Cas9 to cut immediately adjacent to the deletion end points detected in the spontaneous PR mutants (Fig 2A). Analysis of these *cyr1Δ/Δ* mutants that were created to be heterozygous *bcy1Δ*, 270kbΔ, and 590kbΔ in chromosome 2 showed that they recapitulated the phenotypes of the corresponding spontaneous mutants PR13, PR12, and PR18, respectively (Figs 2B–2D and S1). This confirmed that haploinsufficiency of *BCY1* improved growth, and that haploinsufficiency of the genes in the 270 kb region in the left arm of chromosome 2 was sufficient to increase hyphal formation.

Gene mapping strategies to identify the basis for the improved hyphal growth of the PR mutants showed that this effect is due to multiple genetic changes (Figs 2E and S1C). In the

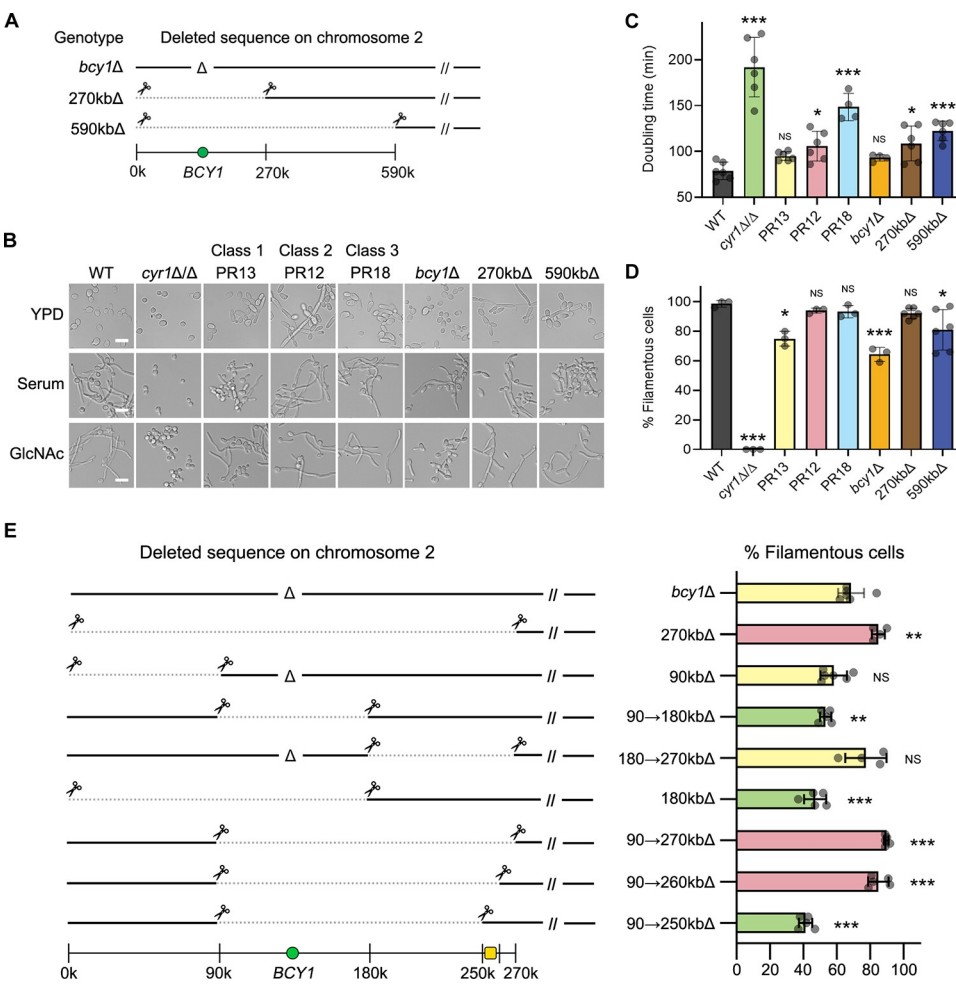

**Fig 2. Gene mapping by CRISPR-Cas9 demonstrates that haploinsufficiency of genes in chromosome 2 improved hyphal induction in *cyr1Δ/Δ* PR mutants.** (A) A schematic diagram showing deleted sequences on chromosome 2. (B) The strains indicated at the top were grown in the liquid medium indicated on the left, and then hyphal induction was assessed microscopically. Cells were grown in liquid medium containing 15% serum or 50 mM GlcNAc to induce hyphal growth at 37˚C for 3 h and then photographed. Scale bar, 10 μm. (C) Doubling times were measured in liquid YPD medium at 30˚C. Shown is the mean ± SD of 6 independent experiments. The doubling time of the strains shown was significantly shorter when compared to the *cyr1Δ/Δ* background ($p < 0.01$). (D) Graph indicating the percent of filamentous cells after growth in GlcNAc medium described in panel B. Shown is the mean ± SD of at least 3 independent experiments with at least 100 cells counted for each condition. (E) Gene mapping by CRISPR-Cas9 identified a 10-kb region (yellow square) that is involved in the filamentous phenotype. The left panel shows deleted sequences on chromosome 2. The right panel shows the percent of filamentous cells in liquid GlcNAc medium at 37˚C; green, weak hyphal induction; yellow, intermediate hyphal induction; pink, strong hyphal induction. (A and E), Note that deletions are heterozygous; the cells retain a wild-type version of chromosome 2. Numbers and symbols indicate gene deletion (Δ), CRISPR cut site (scissors), large genomic deletion (dotted line), *BCY1* gene (green circles), and chromosomal position (kb). (C and D) Statistical analysis was performed using one-way ANOVA with Dunnett's multiple comparisons test comparing the strains with the WT or parental strain; [NS] $p > 0.05$, [*] $p < 0.05$, [**] $p < 0.01$, [***] $p < 0.001$.

first approach, smaller 90 kb sections of the 270 kb region were deleted from one copy of chromosome 2 in a *bcy1Δ cyr1Δ/Δ* strain. Surprisingly, these mutants were not better at producing filamentous hyphal cells during GlcNAc induction (Fig 2E). However, the basal level of filamentous growth was lower in the 90kb deletion mutant, unchanged in 90kb→180kb deletion mutant, and elevated in 180kb→270kb deletion mutant compared to *bcy1Δ cyr1Δ/Δ* strain

(S1C Fig). This indicates that filamentation phenotype is affected by multiple genes in the critical region of chromosome 2.

We used the 90kb→270kb heterozygous deletion mutant as a starting point to map the right end of the genes in this region that impact filamentous growth. A smaller deletion mutant of chromosome 2 from 90kb→260kb was similar to the 90kb→270kb deletion, which partially narrowed the key region. When the right end of the critical region was deleted by another 10kb, the 90kb→250kb deletion mutant did not exhibit better hyphal growth in the presence of GlcNAc. Haploinsufficiency of the 10-kb region was therefore necessary to improve hyphal induction in this strain (although it was not sufficient on its own).

Further mapping by deletion of the genes in this 10-kb region identified *SRB9* and *SPT5* as contributing to the improved hyphal phenotype (S3 Fig). *SRB9* encodes a subunit of the RNA polymerase II Mediator complex and *SPT5* encodes a transcription elongation factor complex subunit. Interestingly, the double heterozygous mutation of *SRB9* and *SPT5* in the 90kb→250kb deletion strain promoted improved hyphal growth, while the single heterozygous mutations of each gene did not. According to previous analyses of haploinsufficiency genetic interactions [43,44], this suggest that the two genes function in the same pathway, consistent with both *SRB9* and *SPT5* encoding subunits of the general transcription machinery. Interestingly, mutation of a different subunit of the Mediator complex (*SSN3*) was identified in a previous study because it restored the ability to form hyphae to a *cph1Δ/Δ efg1Δ/Δ* mutant strain of *C. albicans* that lacks two key hyphal TFs [25]. Thus, the gene mapping results indicate that global transcriptional regulators are important for creating a physiological state in the cell that is more conducive to induction of the switch of hyphal morphogenesis [25,45].

## Adenylyl cyclase is not necessary for transcriptional induction of hyphal regulated genes

Based in part on previous studies of *cyr1Δ/Δ* mutant strains, adenylyl cyclase was thought to be central to inducing the expression of a variety of genes, including virulence factors such as adhesins and hyphal-specific genes that promote filamentous growth [2,3,13,15,16]. To examine this role of cAMP signaling, we performed RNA sequencing (RNA-seq) analysis of the WT and PR mutants before and after stimulation with the hyphal inducer GlcNAc (Fig 3A and S2 Data). Heatmap clustering and principal component analysis (PCA) of the normalized RNA-seq dataset revealed that WT and PRs showed similar patterns in transcriptomes, indicating that adenylyl cyclase is not required to induce the broad range of hyphal genes (Figs 3B and S4). In contrast, the *cyr1Δ/Δ* mutant clustered separately due to its expected defect in regulating hyphal genes (Fig 3B). Data for a subset of hyphal genes, as well as control genes are shown in Fig 3C. These data show that the transcriptomic profile of class 1 mutant PR13 (*bcy1*\*/*BCY1 cyr1Δ/Δ*) was similar to the WT in having a low basal level of hyphal gene expression that was induced by hyphal stimulation. However, class 2 mutant PR12 (270kbΔ *cyr1Δ/Δ*) was distinct in that it expressed a higher basal level of hyphal genes even in the absence of the hyphal inducer GlcNAc (Fig 3C), which correlated with a higher basal level of filamentous cell morphology (Fig 1B). The PR12 transcriptome was also distinct as the levels of the genes in the 270-kb region were around 2-fold less than the WT level due to the haploinsufficiency (see S2 Data). GlcNAc catabolic genes were highly expressed in all the strains tested, including *cyr1Δ/Δ* (Fig 3C), which is consistent with previous studies showing that their regulation is cAMP independent [46,47]. Altogether, these data indicate that an appropriate basal level of PKA activity is needed for cells to respond transcriptionally to a hyphal inducer, rather than a requirement for stimulation of Cyr1 to produce cAMP.

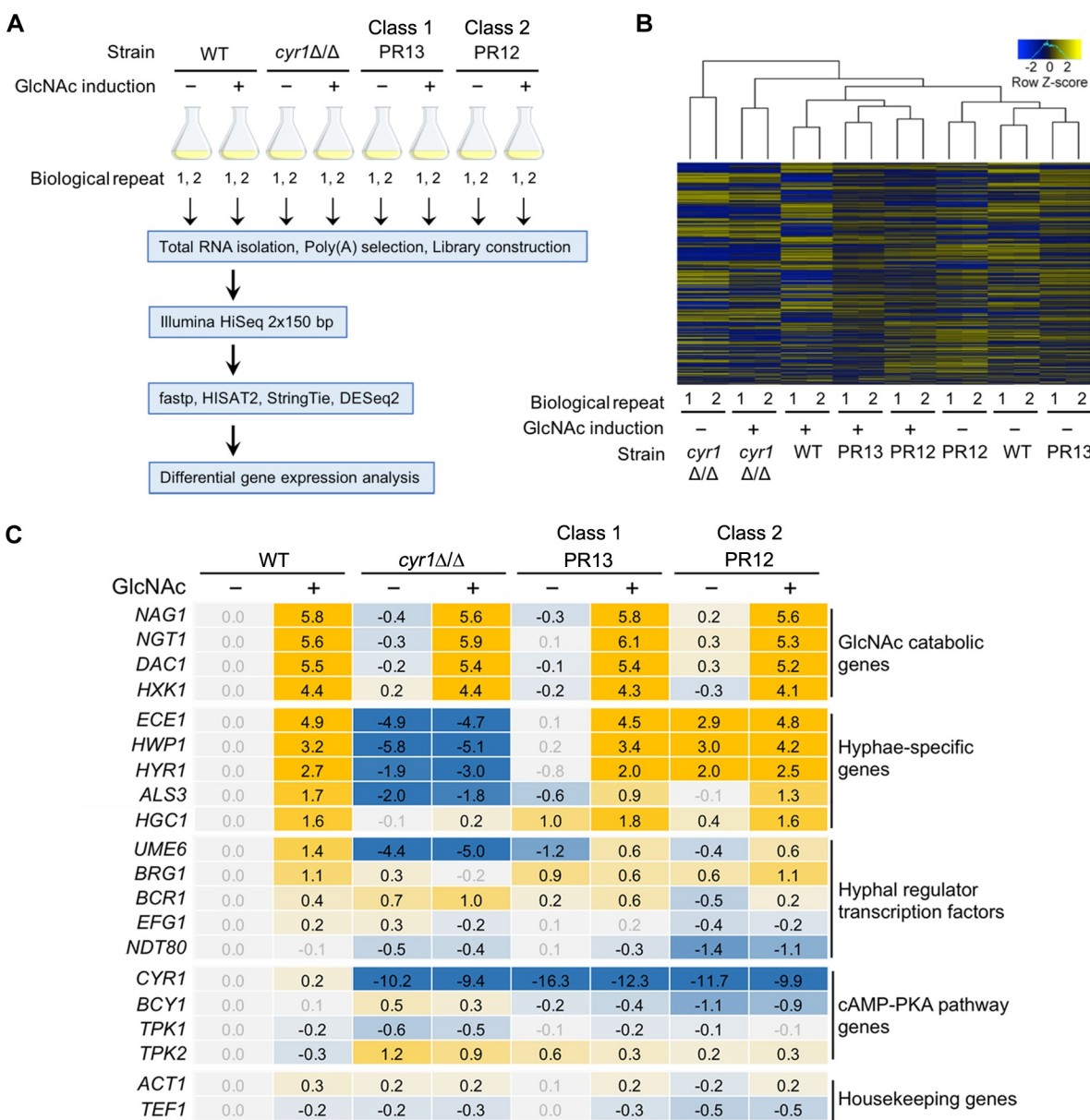

**Fig 3. Adenylyl cyclase is not necessary for transcriptional regulation of hyphal-induced genes.** (A) Experimental scheme of transcriptomic analysis. Cells were grown at 37°C in liquid galactose medium and then 50 mM GlcNAc was added for 2 h to induce hyphae. There were 2 biologically independent replicates for each condition. (B) Cluster analysis of differentially expressed genes (DEGs). (C) Summary of differential expression analysis. The numbers in colored boxes are log$_2$ difference of transcript levels compared to the wild-type −GlcNAc condition. Grey numbers indicate the difference is not significantly different (adjusted p > 0.1).

## Quantitative phosphoproteomics identified protein kinases required for normal hyphal growth

To better understand how the switch to hyphal morphogenesis is regulated, we compared protein production and phosphorylation during hyphal induction. WT and PR13 strains were treated with or without GlcNAc to form hyphae (Fig 4A), and then a label-free mass spectrometry approach (LC-MS/MS) was used to detect 3,434 proteins and 17,580 phosphopeptides (S3 Data). To differentiate experimental variances from biological alterations, we compared the

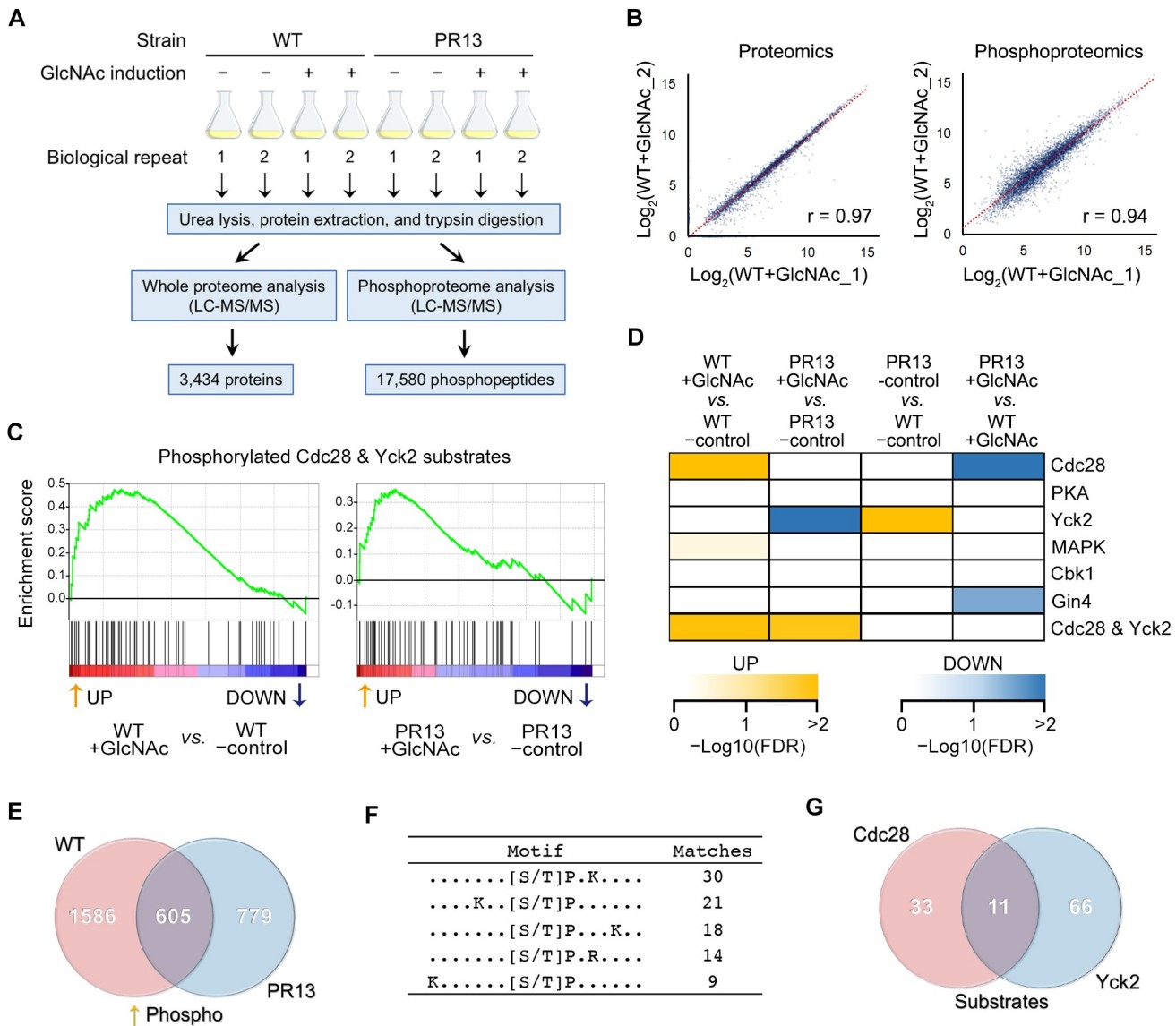

**Fig 4. Quantitative phosphoproteomics detected activity of two key protein kinases in hyphal induction.** (A) Experimental scheme of parallel proteomic and phosphoproteomic analyses. Cells were grown at 37°C in liquid galactose medium and then 50 mM GlcNAc was added for 2 h to induce hyphae. For phosphorylation analysis, phosphopeptides were enriched by polymer-based metal-ion affinity capture (PolyMAC). There were 2 biologically independent replicates for each condition. (B) Representative null comparisons of the biological replicates show high reproducibility. We compared the replicate data of WT+GlcNAc_1 and WT+GlcNAc_2 from both whole proteome and phosphoproteome. (C) Representative results of kinase-substrate enrichment analysis. Gene set enrichment analysis (GSEA) algorithm [50] was used to identify significantly enriched or depleted groups of phosphorylated substrates. The potential substrates of both Cdc28 and Yck2 kinases were more phosphorylated (up-regulated) during the hyphal induction in the WT and PR13 cells. (D) The color-coded diagram illustrates changes of the selected kinase activities indicated by the scale bar at the bottom (< 25% false discovery rate [FDR]). Kinase activity inference was based on the collective phosphorylation changes of their identified substrates. (E) Venn diagrams of phosphopeptides upregulated during hyphal induction ($\log_2$ fold change > 0.5). Note that 605 phosphopeptides were upregulated in both WT and PR13 strains. (F) Overrepresented motifs were extracted from the 605 phosphopeptides, using the motif-x algorithm [88] ($p < 10^{-6}$). 44 phosphopeptides had the Cdc28 consensus motif ([S/T]-P-x-K/R) indicating they were potential substrates. (G) Venn diagrams of the upregulated Cdc28 substrates ([S/T]-P-x-K/R) or Yck2 substrates (S/T-x-x-[S/T]). 11 phosphopeptides were potential substrates of both Cdc28 and Yck2 kinases.

replicate data of WT cells treated with GlcNAc from both the whole proteome and the phosphoproteome. The null comparisons of the biological replicates showed high correlations (r > 0.9) (Fig 4B). We first examined the phosphorylation status of Efg1, since genetic studies implicated this TF as being a target of PKA and Cdc28 kinases on positions T179 and T206

positions, respectively [17,34]. However, our phosphoproteome data did not detect phosphorylation at these sites in Efg1, consistent with the reports of Willger et al. [45] and Cao et al. [48] (see S3 Data). Thus, three completely independent MS studies have not detected phosphorylation of Efg1 which was implicated from genetic evidence. Interestingly, during hyphal induction Efg1 was dephosphorylated in the C-terminal prion-like domain (PrLD), which is a region of the Efg1 protein that is involved in the assembly of TF complexes by formation of phase-separated condensates [49]. Altogether, these results indicate that models for role of Efg1 regulation by phosphorylation during hyphal induction should be revised (see Discussion).

We next used Gene Set Enrichment Analysis [50] to infer the activity of specific protein kinases. The algorithm was based on the levels of phosphorylated peptides containing amino acid sequence motifs characteristic of the substrate specificity of known protein kinases (Fig 4C). Strikingly, in contrast to the current models [2,3,5], this analysis revealed PKA activity was not significantly different before and after hyphal induction in the WT or PR13 strains (Fig 4D). Furthermore, control studies showed that the levels of the PKA subunits Bcy1 and Tpk2 remained constant during hyphal induction (S2 Fig). Thus, there was no evidence to support the stimulation of PKA in the PR mutants during hyphal induction. In contrast, Cdc28 kinase activity was increased during hyphal induction in the WT strain (Fig 4D), consistent with previous studies implicating this cyclin dependent kinase in hyphal regulation [27,28,34,51]. To examine the role of Cdc28 further, we deleted the hyphal-induced cyclin *HGC1*. The *hgc1Δ/Δ* mutation caused a stronger hyphal defect in the PR13 mutant compared to the WT control cells (S5A and S5B Fig), indicating that PR13 cells are more reliant on cyclin-dependent activity for hyphal induction than are the WT cells. Despite of the abnormal morphology, the *hgc1Δ/Δ* mutant produced a similar number of filamentous cells as the WT control strain (~95%). However, the PR13 *hgc1Δ/Δ* mutant produced a lower level of filamentous cells (36%) compared to the PR13 strain (61%).

Interestingly, many of the potential Cdc28 phosphosites also contained a second motif indicating they are targets for Yck2, a member of the casein kinase I family (Fig 4E–4G). Recent studies in *C. albicans* have characterized the role of Yck2 in modulating morphogenesis, virulence, fungal cell-wall stress responses, and resistance to clinical antifungal drugs [52–54]. Deletion of *YCK2* strongly affected hyphal growth in both the WT and PR13 strains (S5A and S5B Fig). The *yck2Δ/Δ* cells displayed an elongated cell morphology, but did not switch to a true hyphal morphology in the presence of the hyphal inducers serum or GlcNAc. Interestingly, both kinases are predicted to phosphorylate 11 sites in 8 proteins that are all thought to have a role in polarized growth (S2 Table). We created phospho-site mutations of *BNI1* and *MOB2* because these genes are required for normal hyphal growth [32,55]. However, mutation of the individual phosphorylation sites did not cause a detectable hyphal defect, suggesting that mutations of additional target proteins may be required to cause a strong defect in hyphal morphogenesis (S5C Fig).

## Hyphal stimulation induces translation of TFs that regulate morphological switching

The transcriptomic and proteomic data sets were examined to determine how the abundance of mRNAs and their corresponding proteins correlated during the switch to hyphal morphogenesis (Fig 5A). There was a relatively weak correlation (Pearson correlation coefficient, $r = 0.39$) between mRNA and protein abundances in *C. albicans* (Fig 5B). Typical Pearson correlation coefficients ($r$) for mammalian cells are about 0.6 and for *S. cerevisiae* it is about 0.7 [56]. This indicated that *C. albicans* has dynamic post-transcriptional regulatory mechanisms.

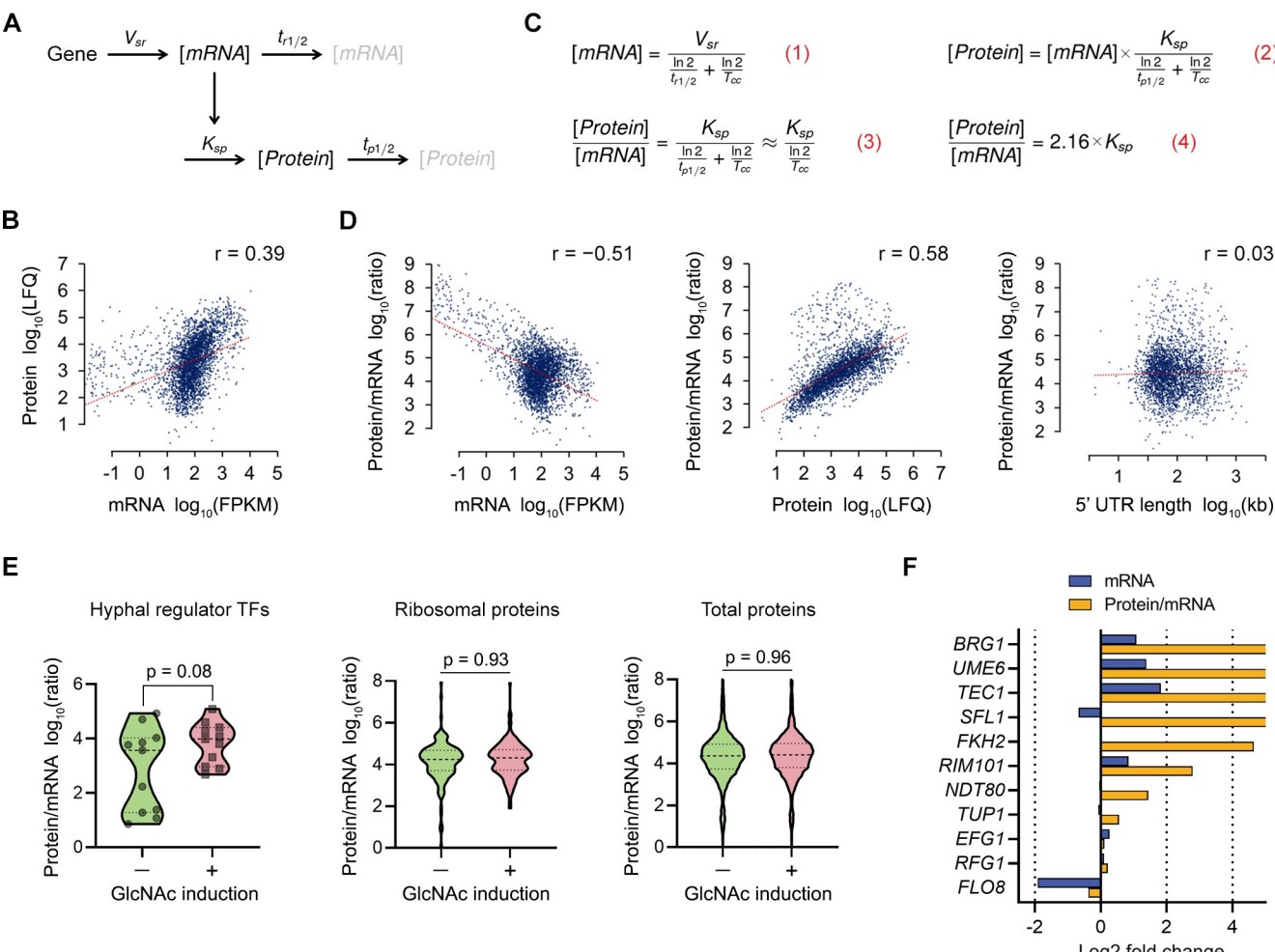

**Fig 5. Integrated transcriptomic and proteomic analyses indicate that hyphal stimulation induces translation of hyphal regulator TFs.** (A) A basic gene expression model with key parameters (adapted from REF. [56,92]). The mRNA is transcribed with rate $V_{sr}$ and degraded with a half-life represented by $t_{r1/2}$. The protein is translated proportionally to the mRNA abundance with the rate constant $K_{sp}$ and degraded with a half-life of $t_{p1/2}$. (B) An across-gene correlation analysis comparing estimates of absolute mRNA abundance (expressed in fragments per kilobase of transcript per million mapped reads (FPKM)) to protein abundance (expressed as label free quantitation (LFQ)) in exponentially growing WT cells. r, Pearson correlation coefficient. (C) Mathematical expression of mRNA (Eq 1) and protein abundances (Eq 2) as a function of key gene expression parameters, including cell doubling time $T_{cc}$, as detailed in REF. [92]. Rearranging Eq 2 yields the dependency of protein-mRNA abundance ratio on the translation rate constant $K_{sp}$, cell doubling time $T_{cc}$, and protein half-life $t_{p1/2}$ (Eq 3). We neglected protein degradation assuming that protein replacement is generally driven by dilution due to cell division in exponentially growing *C. albicans* cells ($t_{p1/2} \gg T_{cc}$). Substituting the cell doubling time ($T_{cc}$ = 1.5 h in *C. albicans*) yields Eq 4. Therefore, we can approximate the translation rate with the protein-to-mRNA level ratio. (D) Across-gene correlation analyses comparing protein-to-mRNA ratio versus mRNA abundance, protein abundance, and 5' UTR length (from left to right). r, Pearson correlation coefficient. (E) Protein-to-mRNA ratios of 11 selected hyphal regulator transcription factors (TFs) during GlcNAc induction in WT. The protein-to-mRNA ratios of the 5 hyphal regulators were two orders of magnitude lower than the median in −GlcNAc control, indicating their translation rate was very low before hyphal induction. The p-value was calculated using two-sided unpaired t-test (hyphal regulator TFs) and z-test (ribosomal proteins and total proteins). (F) The relative change in protein-to-mRNA ratio for the selected 11 hyphal regulators is shown in yellow, relative changes in mRNA expression are shown in blue. The protein-to-mRNA ratio of *UME6*, *SFL1*, *BRG1*, *TEC1*, and *FKH2* increased dramatically (log2 fold change > 4) during hyphal induction while mRNA levels did not.

Since protein levels are a function of mRNA abundance, translation rate, and protein turnover, we approximated the translation rate with the protein/mRNA level ratio (this approximation neglects protein degradation, see Fig 5C). Integration of RNA-seq and label-free LC-MS/MS data gave an opportunity to estimate the translation rate of 3,267 genes (S4 Data). The translation rate spanned over five orders of magnitude across genes (Fig 5D). There was a negative correlation between mRNA abundance and translation rate, while there was a positive

correlation between protein abundance and translation rate. Although it has been reported that 5' UTR length plays an important role in translation efficiency [57–59], there was no correlation between 5' UTR length and translation rate in our analysis.

We next examined whether the translation rate changed for TFs that are known to regulate hyphal morphogenesis. As a group, the predicted translation rate of the 11 hyphal regulator TFs did not change significantly after hyphal induction (Fig 5E). However, there was a huge variation in the translation rates of the hyphal regulator TFs before hyphal stimulation and some of them were very low. Strikingly, translation rates of these TFs increased dramatically during hyphal growth while mRNA levels did not change much compared to the translation rate (Figs 5F and S6). The mean translation rates of ribosomal proteins and total proteins were similar, indicating selective translation regulation of the TFs. The Ume6, Sfl1, Brg1, Tec1, and Fkh2 proteins were not detected in the uninduced cells while their mRNA levels were similar to the median of total mRNAs (S4 Data). Using ribosome profiling, Mundodi et al. [60] reported that many genes involved in hyphal growth showed reduced translational efficiency (ribosome density) in contrast to our finding. However, due to biological factors such as different translation elongation speed, ribosome queuing and the amino acid composition of the nascent peptide chain, as well as technical issues, such as data normalization issues, ribosome densities as measured by ribosome profiling do not necessarily reflect the protein output of the ribosome [56,61]. Further work will be required to confirm whether these effects are due to translation or another mechanism, such as increased protein stability [62,63]. However, the results demonstrate that a post-transcriptional mechanism regulates key transcription factors that are important for hyphal morphogenesis.

## Discussion

The ability of *C. albicans* to switch between budding and hyphal morphologies contributes to virulence by promoting invasive growth, escape from immune cells, and biofilm formation [1–6]. The prevailing model is that the Cyr1 adenylyl cyclase acts as a master regulator by integrating signals from different hyphal inducers [3,5,9–11]. Although stimulation of Cyr1 to increase cAMP levels can trigger hyphal growth, it is not clear that the failure of *cyr1*Δ/Δ cells to form hyphae indicates that Cyr1 plays an essential role. The phenotype of *cyr1*Δ/Δ cells is complicated by the fact that they grow slowly and have an abnormal physiology [12,13]. The identification of faster growing *cyr1*Δ/Δ PR mutants that form filamentous cells in response to inducers including serum, GlcNAc, and alkaline pH indicated the presence of alternative pathways that act in the absence of Cyr1 and cAMP [14]. In this study we demonstrated that the improved growth of the PR mutants is due to mutation or deletion of one of the two copies of *BCY1*, which encodes the negative regulatory subunit of PKA (Figs 1–3 and S2). This is expected to increase PKA activity and confirms that an appropriate basal level of PKA activity enables cells to grow better in the absence of cAMP. These results are also interesting in that genomic analysis revealed that the different classes of PR mutants showed distinct mechanisms for mutation of *BCY1* or deletion of parts of the left arm of chromosome 2 (S1 Table), which underscore how the plasticity of the *C. albicans* genome enables it to adapt to stress from mutations or antifungal drugs [37,38,64,65]. It was also significant that the different chromosomal changes improved growth to different extents, and that the degree of improved growth for the different PR mutant classes did not correlate with their ability to form hyphae (Fig 1). This indicated that the additional genes affected by the changes in chromosome 2 also contributed to the ability of PR mutants to form hyphae.

Current models based on genetic evidence predict that PKA phosphorylates the Efg1 TF to stimulate it to induce expression of hyphal genes [2,3,11,13,15,16]. However, we failed to

detect phosphorylation of Efg1 in WT or PR13 cells, consistent with two previous phosphopro-teomic studies [45,48]. In spite of this, the PR mutants were not only able to form hyphae, but they also induced the expression of the expected set of hyphal-associated genes that includes many virulence factors (Fig 3C). A further limitation of the current models is that several studies now question the role of hyphal-induced genes in promoting hyphal growth. For example, the common core set of genes stimulated by a group of different hyphal inducers in *C. albicans* do not account for the transition to hyphal growth [19]. In addition, under certain conditions GlcNAc can stimulate hyphae without induction of hyphal-specific genes [20]. There also appears to be a lack of TF specificity, as some hyphal regulatory TFs can be bypassed under special environmental conditions [21], or by altered expression of the Cak1 protein kinase [23] or the absence of a variant histone H3 [24]. In addition, hyphal growth was restored in a *cph1*Δ/Δ *efg1*Δ/Δ mutant by a mutation in the *SSN3* subunit of the Mediator complex [25]. Consistent with this, we found that haploinsufficiency of two global transcriptional regulators, *SRB9* (a component of Mediator) and *SPT5*, contributed to the improved filamentous growth of the Class 2 mutants (S3 Fig). Altogether, these data suggest a new model that TFs are important for creating the proper physiological environment for cells to undergo hyphal growth, rather than inducing specific target genes that are directly involved in promoting filamentous morphogenesis (Fig 6).

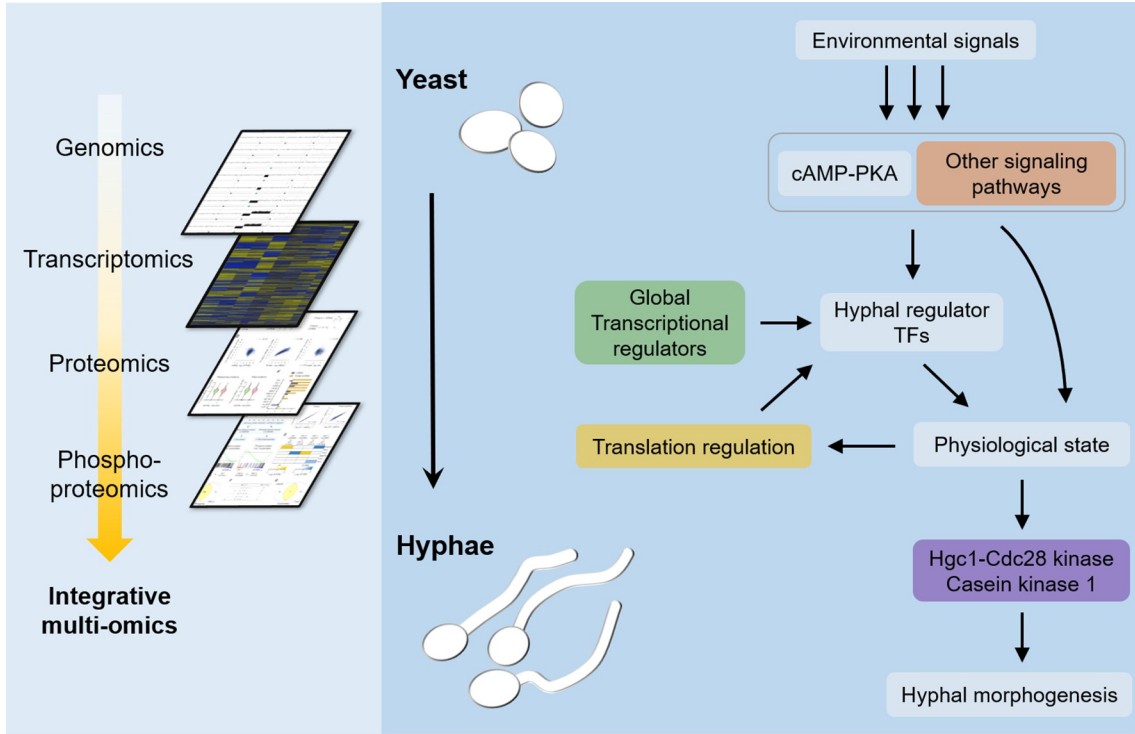

**Fig 6. Integrative multi-omics profiling revealed new pathways that induce *C. albicans* hyphal formation in a cAMP-independent manner.** New insights into the mechanisms of hyphal morphogenesis were obtained by performing integrative multi-omics profiling in *C. albicans*. Genetic changes in PR mutants were identified by performing genome analysis, gene-mapping by CRISPR-Cas9, and transcriptome analysis. These studies showed that gene dosage effects on *BCY1* (negative regulator of PKA) and global transcription mechanisms promoted hyphal growth in the absence of cAMP. Integrating transcriptomic and proteomic data revealed that the translation rate of some hyphal regulatory TFs increased dramatically during hyphal induction while their mRNA levels did not, suggesting a special type of translational regulation promotes hyphal growth. Parallel comparisons of protein production and phosphorylation revealed that phosphorylation by Hgc1-Cdc28 kinase and yeast casein kinase 1 (Yck2) increased during hyphal induction, even in the absence of adenylyl cyclase, consistent with genetic studies indicating that these kinases play an important role in hyphal growth. Altogether, these data suggest a new model that various hyphal inducers and TFs are predicted to induce a new physiological state that promotes polarized filamentous growth, in part due to changes in protein phosphorylation.

A perplexing aspect of hyphal induction is that it can be induced by such a broad range of different environmental and nutritional conditions (e.g. 37˚C, alkaline pH, nitrogen levels, GlcNAc, certain amino acids) [2,3]. We suggest that there are many different pathways that can lead to a physiological state that induces hyphal growth, which helps to explain why there does not appear to be a common transcriptional response to different hyphal inducers. However, TFs can play a role in inducing the appropriate state as mutation of *TUP1* or overexpression of *UME6* is sufficient to induce hyphal growth [16,66]. These TF mutants display changes in a wide range of genes that can impact cell physiology. In addition, studies presented in Fig 5 indicate a role for positive feedback in that induction of hyphal growth can lead to increased production of hyphal TFs (Fig 6). This model also helps to explain other data that metabolic pathway mutants can influence hyphal morphogenesis, such as mitochondrial mutants [2,67–69].

A further prediction of the model shown in Fig 6 is that the new physiological state is expected to lead to activation of protein kinases such as Cdc28 and Yck2. These kinases have been shown previously to be important for inducing hyphal growth in WT cells, and our phosphoproteomic data indicate that they are important in the absence of cAMP. In fact, it appears that PR mutants are more reliant on Hgc1-Cdc28 than are WT cells (S5 Fig). An interesting aspect of the phosphoproteomic data is that we identified a subset of proteins which contain peptides that are phosphorylated on sites for both Cdc28 and Yck1. Although mutating single sites in two of these proteins that are implicated in hyphal morphogenesis did not cause a strong phenotype, it is possible that there are synergistic effects of Cdc28 and Yck2 via regulation of a larger set of substrate proteins. Altogether, these results indicate that future studies on hyphal signaling should focus on the roles that post-transcriptional and post-translational regulation play in promoting the changes in the morphogenesis pathways that mediate highly polarized hyphal growth.

## Materials and methods

### Strains, culture conditions, and hyphal analysis

The *C. albicans* strains used in this study are described in S3 Table. *C. albicans* strains were streaked fresh from storage at -80˚C onto YPD (1% yeast extract, 2% peptone, 2% glucose) agar plates and then grown in rich YPD medium or in complete synthetic medium (SC) made with yeast nitrogen base, select amino acids, and uridine. *C. albicans* transformants were selected on YPD plus 200 µg/mL nourseothricin (NAT; Gold Biotechnology). NAT-sensitive derivatives of cells carrying the SAT flipper were then obtained by growing cells on maltose agar medium (1% yeast extract, 2% peptone, 2% maltose) for 2 days, and then several colonies were spread on YPD plates containing 25 µg/mL of NAT and incubated for 2 days at 30˚C [70]. NAT-sensitive cells formed smaller colonies than NAT-resistant parental strains.

Strain doubling times were determined from growth curves in YPD at 30˚C. Fresh overnight cultures were diluted to a final concentration of ~0.05 $OD_{660}$ in 5 mL YPD in test tubes and then incubated at 30˚C on a tube roller. The $OD_{660}$ was measured hourly for ~8 hours until $OD_{660}$ was >1.0. The reported doubling times are the averages of two biological replicates each performed in duplicate.

To analyze hyphal formation in liquid media, cells were grown overnight at 30˚C in YPD, and then inoculated into SC + 50 mM glucose + 10% bovine calf serum, or SC + 50 mM GlcNAc medium. Samples were then incubated at 37˚C for the indicated time and then images were captured using a Zeiss Axio Observer 7 inverted microscope equipped with a 40× objective differential interference contrast (DIC) optics and a Zeiss AxioCam 702 digital camera. We counted both hyphae and pseudohyphae as filaments to make the image analysis consistent. *C. albicans* cells were grown and induced to make hyphae in the same conditions for

transcriptome, proteome, and phosphoproteome analyses. Galactose was used as a carbon source because glucose represses expression of the GlcNAc transporter *NGT1*, which limits hyphal induction [71]. *C. albicans* cells were grown at 37°C to log phase (OD$_{660}$ of 0.2 ~ 0.4) in 450 mL of SC + 50 mM galactose medium. For hyphal induction, 50 mL of pre-warmed 0.5 M GlcNAc solution was added into the culture to make final concentration of 50 mM and the cultures were incubated for 2 h at 37°C. Cells were harvested by centrifugation, washed three times with ice-cold ultrapure water, snap frozen on dry ice, and stored at −80°C. Replicate experiments were conducted independently on different days.

## Strain construction

Deletion mutant strains were created using transient expression of CRISPR-Cas9 to facilitate the genome engineering and limit transformation-induced genomic changes in *C. albicans* [41,72]. The methods were performed essentially as described previously [41,42]. Briefly, *CaCAS9* expression and sgRNA expression cassettes were co-transformed into cells along with a repair template. The *CaCAS9* gene was codon optimized for expression in *C. albicans* [73]. The *CaCAS9* expression cassette was PCR amplified from the plasmid pV1093, which was a kind gift from Dr. Valmik Vyas [73]. The sgRNA expression cassette was constructed through single-joint PCR from the plasmid pV1093 [41]. We used 20-bp target sequences of the sgRNA, as reported previously by Vyas et al. [74] to target Cas9 to make a DNA double strand break at the target site (S4 Table). When deleting large regions of chromosome 2 for gene mapping, two sgRNAs were used to cut each side of the target region (see S1 Text). Repair templates were synthesized using the plasmid pGR-NAT as a template, which contains the *SAT1* flipper cassette [70]. The primers were designed to include 60 to 80 bases of homology to the sequences upstream or downstream from the target region. The oligonucleotides used in this study are listed in S5 Table. PCR was conducted with Ex Taq (TaKaRa Bio, Inc.) in accordance with the CRISPR protocol (S1 Text).

*BNI1* and *MOB2* were mutated to prevent phosphorylation at the indicated sites using the CRISPR-Cas9 method developed by Vyas et al. [73] The 20-bp target sequence of sgRNA was cloned into pV1093 and correct clones were verified by DNA sequencing. The CRISPR expression plasmids were linearized by digestion with *Kpn1* and *Sac1* before transformation for efficient targeting to the *ENO1* locus. Repair templates were generated with two 60-base oligonucleotides containing a 20-base overlap at their 3′ ends centered on the desired site to mutate a phosphorylation site. The repair templates were constructed by PCR primer extension to contain the desired phospho-site mutation as well as a unique restriction site to facilitate identification of appropriate transformants.

PCR products and linearized plasmids for transformation were purified and concentrated using a GeneJET PCR purification kit (Thermo Fisher Scientific, Inc.). Electroporation was used to introduce the DNA into cells following a previously described method [75]. An electrocompetent cell suspension (40 µl) was added to aliquoted DNA, placed in 0.1 cm-gap electroporation cuvettes, and electroporated on a Bio-Rad Gene Pulser at 1.5 kV. One milliliter of 0.5× YPD containing 1 M sorbitol was added immediately to the cuvette, and then the cell mixture was incubated for 3 h at 30°C before plating onto YPD + 200 µg/mL NAT agar. Nat$^r$ transformants were selected, and PCR genotyping of the transformants verified the genome editing. When deleting large regions of chromosome 2 for gene mapping, we confirmed that the deletions are heterozygous. As described in a previous paper [41], two PCR detection strategies were designed to detect the wild-type allele and the deletion allele. More than 10 independent transformants were tested for each deletion and around half of them had both WT and deletion alleles (heterozygous deletion). We did not find any homozygous deletion mutants, presumably due to the essential genes present in these regions of chromosome 2.

## Genome analysis

A 1 μg aliquot of the genomic DNA was prepared using the Illumina TruSeq PCR-free DNA HT sample preparation kit with 450 bp insert size. Intact genomic DNA was sheared using the Covaris sonicator (adaptive focused acoustics), followed by end-repair and bead-based size selection of fragmented molecules. The size selected fragments then had adenines added to the 3' ends of the DNA, Illumina sequence adaptors ligated onto the fragments, followed by PCR amplification and final library QC. A majority of the steps in this process were performed on the Caliper Sci-Clone NGSx workstation (PerkinElmer), a robotics system developed and validated for automated library preparation. The library QC included a measurement of the average size of library fragments using the Agilent BioAnalyzer, estimation of the total concentration of DNA by Pico-Green (ThermoFisher), and a measurement of the yield and efficiency of the adaptor ligation process with a quantitative PCR assay (Kapa) using primers specific to the adaptor sequence. Sequencing was performed on the Illumina MiSeq instrument at $2 \times 150$ bp read length.

Reads were first trimmed and quality filtered using skewer v0.1.127 [76] with the following parameters (-q 20 -Q 20 -l 50 -n yes). Reads were then mapped to the *C. albicans* SC5314 genome (Assembly 21) [77] using BWA mem v0.7.15 [78]. Duplicates were marked and read groups added using the PICARD MarkDuplicates and AddOrReplaceReadGroups programs. Indels were realigned using GATK 3.3 RealignerTargerCreator and IndelRealigner [79]. A set of high-quality variants with variant scores of >100 was identified using GATK 3.3 Haplotype-Caller on all samples simultaneously. These were then set as known sites when performing base quality recalibration using GATK 3.3 PrintReads, AnalyzeCovariates and BaseRecalibrator to produce finished bam files. GATK 3.3 HaplotypeCaller was then used again to call variants across samples to produce a final vcf file. snpEff 4.2 was used to annotate variants using a custom built database based on the *C. albicans* SC5314 genome (Assembly 21) [77].

## Transcriptome analysis

RNA extraction, library preparations, and sequencing reactions were conducted at GENEWIZ, LLC. (South Plainfield, NJ, USA). Total RNA was extracted from fresh frozen cell pellets ($10^8$ cells per sample) using Qiagen RNeasy Plus Universal mini kit following manufacturer's instruction (Qiagen). Extracted RNA samples were quantified using Qubit 2.0 Fluorometer (Life Technologies) and RNA integrity was checked using Agilent TapeStation 4200 (Agilent Technologies). RNA sequencing libraries were prepared using the NEBNext Ultra II RNA Library Prep Kit for Illumina following manufacturer's instructions (NEB). Briefly, mRNAs were first enriched with Oligo(dT) beads. Enriched mRNAs were fragmented for 15 min at 94°C. First strand and second strand cDNAs were subsequently synthesized. cDNA fragments were end repaired and adenylated at 3' ends, and universal adapters were ligated to cDNA fragments, followed by index addition and library enrichment by limited-cycle PCR. The sequencing libraries were validated on the Agilent TapeStation (Agilent Technologies), and quantified by using Qubit 2.0 Fluorometer (Invitrogen) as well as by quantitative PCR (KAPA Biosystems). The sequencing libraries were pooled and clustered on 1 lane of a flowcell. After clustering, the flowcell was loaded on the Illumina HiSeq instrument (4000 or equivalent) according to manufacturer's instructions. The samples were sequenced using a $2 \times 150$bp Paired End (PE) configuration. Image analysis and base calling were conducted by the HiSeq Control Software (HCS). Raw sequence data (.bcl files) generated from Illumina HiSeq was converted into fastq files and de-multiplexed using Illumina's bcl2fastq 2.17 software. One mismatch was allowed for index sequence identification.

We performed quality profiling, adapter trimming, read filtering, and base correction for raw data using an all-in-one FASTQ preprocessor, *fastp* [80]. The high-quality paired-end

reads were mapped to the *C. albicans* SC5314 genome (Assembly 22) [81] using HISAT2 [82]. StringTie [83] was used to assemble the read alignments obtained in the previous step and estimate transcript abundances. Absolute mRNA abundance was expressed as fragments per kilobase of transcript per million mapped reads (FPKM). Differential expression analyses were conducted using DESeq2 [84] package from Bioconductor [85] on R.

## Western blotting

The fresh frozen cell pellets ($10^9$ cells per sample) were lysed using the same volume (~100 μL) of 2× SDS lysis buffer (5% SDS, 20% glycerol, 125 mM Tris-HCl, pH 6.8) supplemented with cOmplete EDTA-free protease inhibitor cocktail (Roche), 10 mM activated sodium orthovanadate, 2.5 mM sodium pyrophosphate, 1 mM β-glycerophosphate, 1% phosphatase inhibitor cocktail 2 (Millipore-Sigma), 1% phosphatase inhibitor cocktail 3 (Millipore-Sigma). Zirconia beads were added, and the cells were mechanically disrupted by 3 rounds of 1-min bead beating and 1-min cooling on ice. The samples were boiled at 95°C for 5 min. After centrifugation at 14,000 rpm for 5 min at 4°C to remove cellular debris, the supernatant was transferred to a new tube and this step was repeated. The protein concentration was measured using a BCA protein assay kit (Thermo Fisher Scientific). 2-Mercaptoethanol and bromophenol blue were added to 5% and 0.002% final concentration, respectively. The samples were boiled at 95°C for 5 min. Equal amounts of proteins were resolved by SDS-PAGE using the Mini-PROTEAN Tetra Cell system (Bio-Rad) and Tris/glycine/SDS running buffer. Subsequently, proteins were transferred to a nitrocellulose membrane (Amersham) using a semi-dry transfer apparatus. The membranes were blocked with 5% nonfat dry milk in TBS-T buffer (20 mM Tris-HCl, pH 7.6, 150 mM NaCl, 0.2% Tween-20) for 30 min and probed with CaBcy1 rabbit polyclonal antibody (GenScript) or CaTpk2 rabbit polyclonal antibody (GenScript), at 1:1,000 dilution in 5% nonfat dry milk in TBS-T buffer plus 0.5% sodium azide for 2 hours at room temperature. The membranes were then washed three times with TBS-T and incubated with an IRDye 800CW goat anti-rabbit IgG secondary antibody (LI-COR Biosciences) diluted at 1:10,000 in TBS-T. The membranes were washed three times with TBS-T, stored in TBS buffer, and visualized by scanning with an Odyssey CLx infrared imaging system (LI-COR Biosciences). For Coomassie stained gels, SDS-PAGE gels were performed as described and then stained in Coomassie Brilliant Blue solution (0.1% Coomassie R-250, 50% methanol, 40% water, 10% glacial acetic acid) overnight. Gels were incubated for 6 h in destaining solution (50% water, 40% methanol, 10% glacial acetic acid), rinsed with water, and analyzed by scanning with an Odyssey CLx infrared imaging system (LI-COR Biosciences). Resulting images were analyzed and quantified using ImageStudio software (LI-COR Biosciences).

## Preparation of samples for mass spectrometry

The frozen cell pellet samples ($5 \times 10^9$ cells per sample) were processed by Tymora Analytical Operations (West Lafayette, IN). For lysis, 200 μL of lysis buffer (8M urea in 50mM Tris-Cl pH 7.5, supplemented with phosphatase inhibitor cocktail 3 [Millipore-Sigma]) was added to each of the pellets and pipetted up and down several time to lyse a portion of the pellet. The samples were incubated for 10 min at 37°C, pulse-sonicated several times with a sonicator probe, and incubated again for 10 min at 37°C. The lysed samples were then centrifuged at 16,000 g for 10 min to remove debris and the supernatant portions collected. BCA assay was carried out to calculate the protein concentration and all samples were normalized by protein amount to 200 μg each. Then 5 mM dithiothreitol was added and the proteins were incubated at 37°C for 15 min to reduce the cysteine residues, and then alkylated by incubation in 15 mM iodoacetamide at room temperature for 45 min in the dark. The samples were diluted 3-fold

with 50 mM triethylammonium bicarbonate and digested with Lys-C (Wako) at 1:100 (w/w) enzyme-to-protein ratio for 3 h at 37˚C. The samples were further diluted 3-fold with 50 mM triethylammonium bicarbonate and trypsin was added to a final 1:50 (w/w) enzyme-to-protein ratio for overnight digestion at 37˚C. After digestion, the samples were acidified with trifluoroacetic acid to a pH < 3 and desalted using Top-Tip C18 tips (Glygen) according to manufacturer's instructions. A 1% portion of each sample was used for direct proteomics analysis and the remainder of each sample was used for phosphopeptide enrichment. The samples were dried completely in a vacuum centrifuge and stored at -80˚C. The 99% portion of each sample was used for phosphopeptides enrichment using PolyMAC Phosphopeptide Enrichment Kit (Tymora Analytical) according to the manufacturer's instructions.

### Liquid chromatography with tandem mass spectrometry (LC-MS/MS) analysis

The full phosphopeptide sample and 1 μg of the peptide sample each was dissolved in 10.5 μl of 0.05% trifluoroacetic acid with 3% (v/v) acetonitrile containing spiked-in indexed Retention Time Standard artificially synthetized peptides (Biognosys). The spiked-in 11-peptides standard mixture was used to account for any variation in retention times and to normalize abundance levels among samples. Ten microliters of each sample were injected into an Ultimate 3000 nano UHPLC system (Thermo Fisher Scientific). Peptides were captured on a 2-cm Acclaim PepMap trap column and separated on a heated 50-cm Acclaim PepMap column (Thermo Fisher Scientific) containing C18 resin. The mobile phase buffer consisted of 0.1% formic acid in ultrapure water (buffer A) with an eluting buffer of 0.1% formic acid in 80% (v/v) acetonitrile (buffer B) run with a linear 90-min gradient of 6–30% buffer B at flow rate of 300 nL/min. The UHPLC was coupled online with a Q-Exactive HF-X mass spectrometer (Thermo Fisher Scientific). The mass spectrometer was operated in the data-dependent mode, in which a full-scan MS (from m/z 375 to 1,500 with the resolution of 60,000) was followed by MS/MS of the 15 most intense ions (30,000 resolution; normalized collision energy—28%; automatic gain control target [AGC] - 2E4, maximum injection time—200 ms; 60 sec exclusion).

### LC-MS data processing

The raw files were searched directly against the *C. albicans* database updated in 2019 with no redundant entries, using Byonic (Protein Metrics) and Sequest search engines loaded into Proteome Discoverer 2.3 software (Thermo Fisher Scientific). MS1 precursor mass tolerance was set at 10 ppm, and MS2 tolerance was set at 20ppm. Search criteria included a static carbamidomethylation of cysteines (+57.0214 Da), and variable modifications of phosphorylation of S, T and Y residues (+79.996 Da), oxidation (+15.9949 Da) on methionine residues and acetylation (+42.011 Da) at N terminus of proteins. Search was performed with full trypsin/P digestion and allowed a maximum of two missed cleavages on the peptides analyzed from the sequence database. The false-discovery rates of proteins and peptides were set at 0.01. All protein and peptide identifications were grouped and any redundant entries were removed. Only unique peptides and unique master proteins were reported.

### Label-free quantitation analysis

All data were quantified using the label-free quantitation node of Precursor Ions Quantifier through the Proteome Discoverer v2.3 (Thermo Fisher Scientific). For the quantification of proteomic and phosphoproteomic data, the intensities of peptides were extracted with initial precursor mass tolerance set at 10 ppm, minimum number of isotope peaks as 2, maximum

ΔRT of isotope pattern multiplets– 0.2 min, PSM confidence FDR of 0.01, with hypothesis test of ANOVA, maximum RT shift of 5 min, pairwise ratio-based ratio calculation, and 100 as the maximum allowed fold change. The abundance levels of all peptides/phosphopeptides and proteins/phosphoproteins were normalized to the spiked-in internal iRT standard. For calculations of fold-change between the groups of proteins, total protein or phosphoprotein abundance values were added together and the ratios of these sums were used to compare proteins within different samples. When the protein abundance could not be determined, pseudo value (1) was given for logarithm.

## Phosphoproteomic data analysis

Phosphopeptide abundance values were normalized by protein abundance values [86,87]. Fifteen-amino acid sequences around the phosphorylation site was extracted from the phosphoproteome using a custom Microsoft Excel spreadsheet. rmotifx [88] package on R was used to find overrepresented patterns from the 15-aa sequence set. Potential substrates of 6 kinases (Cdc28, PKA, Yck2, MAPK, Cbk1, and Gin4) were predicted from the phosphoproteome based on their consensus phosphorylation motifs. The following consensus motifs were retrieved from the Scansite 4.0 database [89]; Cdc28 ([S/T]-P-x-K/R); PKA (R/K-R/K-x-[S/T]); Yck2 ([S/T]-x-x-[S/T]); MAPK (P-x-[S/T]-P); Cbk1 (H-x-K/R-x-x-[S/T] or H-x-x-K/R-x-[S/T]); Gin4 (R-S-x-[S/T]). We next utilized a gene set enrichment analysis algorithm, GSEA [50,90], to infer kinase activity based on the substrate phosphorylation levels from the phosphoproteome.

## Supporting information

**S1 Fig. Hyphal induction rate of the *C. albicans* strains.** (A) The plot shows the percent of filamentous cells in YPD medium at 30˚C (-control). (B) Graph indicating the percent of filamentous cells after growth in GlcNAc medium. Cells were grown in liquid medium containing 50 mM GlcNAc to induce hyphal growth at 37˚C for 2 h and then filamentous cells were counted. (C) The percent of filamentous cells in YPD medium at 30˚C; green, weak filamentation; yellow, intermediate filamentation; pink, strong filamentation. Deletions indicated on the left are on chromosome 2 and they are heterozygous; the cells retain a wild-type version of chromosome 2. (A, B, and C) Shown is the mean ± SD of at least 3 independent experiments with at least 100 cells counted for each condition. Statistical analysis was performed using one-way ANOVA with Dunnett's multiple comparisons test comparing the strains with the WT or parental strain; [NS] $p > 0.05$, [**] $p < 0.01$, [***] $p < 0.001$.
(TIF)

**S2 Fig. Expression levels of PKA subunits during hyphal induction.** (A) Western blot detection of the negative regulatory subunit (Bcy1) and the catalytic subunit (Tpk2) of PKA. Cells were grown at 37˚C in liquid galactose medium and then 50 mM GlcNAc was added for 2 h to induce hyphae. The sizes of the protein standards (kDa) are indicated on the right of each blot. Images shown are representative of three independent experiments. (B-D) Relative levels of Bcy1 (B), Tpk2 (C), and Tpk2/Bcy1 ratio (D) compared to the WT 0-h samples. Shown is the mean ± SD of 3 independent experiments. Expression levels were normalized to total proteins on Coomassie-stained gels. Statistical analysis was performed using one-way ANOVA with Dunnett's multiple comparisons test comparing the strains with the WT; [NS] $p > 0.05$, [*] $p < 0.05$, [**] $p < 0.01$, [***] $p < 0.001$.
(TIF)

**S3 Fig. Gene-mapping analysis of the 10-kb region of chromosome 2 identified a role for *SRB9* and *SPT5* in hyphal induction in the PR mutants.** The single, double, and triple heterozygous deletion mutants of *SRB9*, *SPT5*, and *SSL1* were created in KM20 strain background (*Chr2L 90kb→250kbΔ bcy1Δ/BCY1 cyr1Δ/Δ*). The plot shows the percent of filamentous cells in GlcNAc medium. Cells were grown in liquid medium containing 50 mM GlcNAc to induce hyphal growth at 37˚C for 2 h and then filamentous cells were counted.; yellow, intermediate hyphal induction; pink, strong hyphal induction. Statistical analysis was performed using one-way ANOVA with Dunnett's multiple comparisons test comparing the strains with the parental strain; [NS] $p > 0.01$, [***] $p < 0.001$.
(TIF)

**S4 Fig. Correlation analysis and principal component analysis (PCA) of the normalized RNA-seq dataset.** (A) Representative null comparisons of the biological replicates show very high reproducibility ($r ≈ 1.00$) in the RNA-seq dataset. Each dot represents the transcript level of individual gene in the scatter plots. We compared the biological replicates of WT−control and WT+GlcNAc. rLog, regularized log transformation. (B) PCA plot shows clusters of biological replicates based on their similarity in transcriptome.
(TIF)

**S5 Fig. Deletion of Cdc28 cyclin (*HGC1*) and casein kinase 1 (*YCK2*) disrupt normal hyphal growth.** (A) Deletion of Cdc28 cyclin (*HGC1*) and casein kinase 1 (*YCK2*) disrupt normal hyphal growth in WT and PR13 backgrounds. (B) The plot shows the percent of filamentous cells in YPD medium at 30˚C (-control) and after 2-h growth in GlcNAc medium at 37˚C (+GlcNAc). Shown is the mean ± SD of at least 3 independent experiments with at least 100 cells counted for each condition. Statistical analysis was performed using one-way ANOVA with Dunnett's multiple comparisons test comparing the strains with the WT or parental strain; [ns] $p > 0.05$, [****] $p < 0.0001$. (C) Phospho-mutants of Mob2 and Bni1 did not show an obvious defect in hyphal growth. (A and C) The strains indicated at the top were grown in the liquid medium indicated on the left, and then hyphal induction was assessed microscopically. Cells were grown in liquid medium containing 15% serum or 50 mM N-acetylglucosamine (GlcNAc) to induce hyphal growth. Cells were incubated at 37˚C for 2 h and then photographed. Scale bar, 10 μm.
(TIF)

**S6 Fig. Protein-to-mRNA ratios of hyphal regulator TFs during GlcNAc induction in PR13.** The relative change in protein-to-mRNA ratio for the selected 11 hyphal regulators TFs is shown in yellow, relative changes in mRNA expression are shown in blue. The protein-to-mRNA ratio of *RIM101* and *RFG1* increased dramatically (log2 fold change > 4) during hyphal induction while mRNA levels did not.
(TIF)

**S1 Table. Genome analysis summary.** [a] 23-bp telomere repeat sequence, CACCAAGAAGT-TAGACATCCGTA.
(DOCX)

**S2 Table. Potential phosphorylation substrates of Cdc28 and Yck2 during hyphal induction.** [a] Descriptions and deletion phenotypes were obtained from *Candida* Genome Database.
(DOCX)

**S3 Table. *C. albicans* strains used in this study.**
(DOCX)

**S4 Table. Target sequences of the sgRNAs.**
(DOCX)

**S5 Table. The oligonucleotides used in this study.**
(DOCX)

**S1 Data. SNP analysis of the PR mutants.**
(XLSX)

**S2 Data. RNA-seq data.**
(XLSX)

**S3 Data. Quantitative phosphoproteomic data.**
(XLSX)

**S4 Data. Whole-proteomic data and the protein-mRNA abundance correlation.**
(XLSX)

**S1 Text. Protocol for transient CRISPR-Cas9 system.**
(DOCX)

## Acknowledgments

We thank the members of our laboratories for their helpful comments on the manuscript. We are grateful to Jihye An for help with microscopic image analysis and Kyunghye Min for assistance with illustrations.

## Author Contributions

**Conceptualization:** Kyunghun Min, Haoyu Si, James B. Konopka.

**Formal analysis:** Kyunghun Min, Krishna R. Veeramah, John D. Haley, James B. Konopka.

**Investigation:** Kyunghun Min, Thomas F. Jannace, Haoyu Si, Krishna R. Veeramah, John D. Haley.

**Methodology:** Kyunghun Min, James B. Konopka.

**Supervision:** James B. Konopka.

**Visualization:** James B. Konopka.

**Writing – original draft:** Kyunghun Min, James B. Konopka.

**Writing – review & editing:** Kyunghun Min, James B. Konopka.

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
