## [Decision Letter · Decision Letter 0]

4 May 2021

Dear Dr. Konopka,

Thank you very much for submitting your manuscript "Integrative multi-omics profiling reveals cAMP-independent mechanisms regulating hyphal morphogenesis in Candida albicans" for consideration at PLOS Pathogens. As with all papers reviewed by the journal, your manuscript was reviewed by members of the editorial board and by several independent reviewers. In light of the reviews (below this email), we would like to invite the resubmission of a significantly-revised version that takes into account the reviewers' comments.

One main aspect of the study should be better developed to enhance its novelty and clarify an apparent discrepancy with the previous related report from your lab (Parrino et al 2017). The results of the present investigation show that increasing PKA activity, which is reduced when inhibition by the regulatory subunit Bcy1 cannot be relieved in the absence of cAMP, bypasses the requirement for Cyr1. This has occurred in all pseudorevertants analysed in this study, which have only one functional BCY1 allele. So these are classical suppressor mutants in a downstream component of the Cyr1-cAMP pathway. As pointed out by reviewer 2, in that respect the results confirm the established model, that a major role of Cyr1 is to allow PKA activity, and as such do not present a novel paradigm. However, in the previous study it was reported that none of three PR mutants contained mutations in BCY1, although one of them, PR2, apparently was included again in the present investigation and was now found to have lost one of the BCY1 alleles (which are polymorphic in this strain background, so this should have been apparent in the previous analysis). Is it possible that the other previously identified mutants PR1 and PR3 also contain alterations in BCY1 (unrecognised in that study)? Or do these mutants indeed contain suppressor mutations that bypass the requirement for Cyr1 in a way that does not restore PKA activity? This should clarified (see also comments by reviewer 3), and PKA-independent cyr1 suppressors may provide insights into a novel pathway. An aspect that is indeed new (but was already evident in the paper by Parrino et al) is that some signals that induce hyphal growth and were thought to act on Cyr1 can in fact stimulate morphogenesis in a Cyr1-independent manner, but it remains unknown how this occurs. As also pointed out by reviewer 1, the significance of the study would be strongly enhanced if the authors use their multi-omics data to reveal a mechanism how these signals are sensed and transmitted in a Cyr1-independent way.

The reviewers have also made many additional editorial suggestions, mainly to clarify specific points and to tone down statements and conclusions, which should also be addressed in a revised manuscript.

We cannot make any decision about publication until we have seen the revised manuscript and your response to the reviewers' comments. Your revised manuscript is also likely to be sent to reviewers for further evaluation.

Sincerely,

Joachim Morschhäuser

Associate Editor

PLOS Pathogens

Alex Andrianopoulos

Section Editor

PLOS Pathogens

Kasturi Haldar

Editor-in-Chief

PLOS Pathogens

orcid.org/0000-0001-5065-158X

Michael Malim

Editor-in-Chief

PLOS Pathogens

orcid.org/0000-0002-7699-2064

Reviewer's Responses to Questions

**Part I - Summary**

Reviewer #1: For many years the cyr1 adenylate cyclase has been thought to play a critical role in promoting C. albicans hyphal morphogenesis, an important virulence trait of this major human fungal pathogen. In this study the authors characterize pseudorevertants of the cyr1 mutant which are capable of forming filaments in the absence of cAMP. Using genomics they show that chromosome 2 haploinsufficiency at BCY1, a negative regulatory subunit of PKA can rescue the slow growth defect of the cyr1 mutant and an additional multigenic haploinsufficiency at SRB9 and STP5, two global transcriptional regulators, can rescue the filamentation defect of the cyr1 mutant. Integrating information from transcriptomics, proteomics and phophoproteomics revealed that Cdc28 and Yck1 phosphorylation activities are active during filamentation of the pseudorevertants and that several transcriptional regulators of hyphal growth appear to be post-transcriptionally induced.

Overall, the manuscript is well-written and the data are clearly presented. This study is significant because it provides new information about non-cAMP-dependent mechanisms that drive C. albicans morphogenesis. In addition, the multi-omics data highlight the importance of post-transcriptional mechanisms for this morphological transition. However, there are several weaknesses as well. In particular, the manuscript could be strengthened by additional experiments, based on the multi-omics data, that provide new mechanistic information about the yeast-hyphal transition.

Reviewer #2: In the fungal pathogen Candida albicans, adenylyl cyclase (Cyr1), which catalyzes cAMP synthesis, has long been thought to be essential for hyphal morphogenesis. cAMP activates protein kinase A (PKA) to trigger hyphal development. Several years back, James Konopka’s group isolated cyr1Δ/Δ pseudorevertant (PR) mutants that could form hyphae in the absence of cAMP. The finding was fascinating because it seemed to challenge a well-established model of C. albicans hyphal growth. However, how these PR mutants respond to induction to activate hyphal development was unknown.

As a follow-up study here, the Konopka group isolated additional PR mutants and conducted a comprehensive investigation of the molecular basis for the incredible hyphal growth in the PR mutants. First, they discovered that all the PR mutants shared one common genetic change, albeit by different means: the loss of one copy of BCY1, the negative regulatory subunit of protein kinase (A) from the left arm of chromosome 2. Furthermore, while a point mutation in one copy of BCY1 moderately restored hyphal morphogenesis, multigenic haploinsufficiency resulting from loss of large regions of the left arm of chromosome 2 further improved hyphal growth. Using CRISPR/cas9-mediated truncation of the left arm of Chr 2, they identified a 10-kb region important for improving hyphal growth. Interestingly, this region includes global transcriptional regulators. By RNA-Seq transcriptomic analysis, the authors found that many hyphal-associated genes were induced in the absence of cAMP. This indicates that basal PKA activity is an important prerequisite to induce hyphae, but activation of the cAMP pathway is not needed. The phosphoproteomic analysis further indicated that the cyclin-dependent Cdc28 kinase and the casein kinase Yck2 play vital roles required for the hyphal growth in the PR mutants. Finally, integrating transcriptomic and proteomic data reveals that hyphal stimuli induce increased production of key transcription factors that contribute to polarized morphogenesis.

The significance of this study is multi-fold:

1. It demonstrates the fascinating genome plasticity of C. albicans in adaptation to harmful mutations to improve fitness (growth and hyphal development).

2. It indicates the presence of yet unknown signaling pathway that activates the hyphal development in the absence of cAMP.

3. Differential translation of transcription factors could be an important mechanism that contributes to hyphal growth.

The findings presented here open up exciting opportunities for future investigations.

I suggest that the authors modify some of their statements, which are sweeping and overly assertive.

Reviewer #3: The manuscript by Min et al. provides a stimulating analysis of how Candida albicans strains that are deficient in cAMP signaling are still able to undergo hyphal formation in response to certain environmental cues. This is a very important area of research given that filamentation is critical for C. albicans to be pathogenic in multiple infection models. The work is generally of high quality and contains a number of interesting observations and complementary experimental techniques. The fact that hyphae formation (and response to environmental cues) can occur without cAMP/CYR1 continues to be intriguing and indicates much still remains to be learnt about mechanisms of filamentation in this species. Overall, there was much to appreciate in the paper, although I had a number of questions / concerns that, if addressed, would substantially strengthen the paper.

**Part II – Major Issues: Key Experiments Required for Acceptance**

Reviewer #1: 1. The multi-omics experiments are not accompanied by additional experiments providing independent experimental validation (eg: qRT-PCR and Westerns). These experiments are important to include as controls.

2. Hgc1 and Yck1 have previously been shown to play important roles in C. albicans filamentous growth. The manuscript could be significantly strengthened by the demonstration that novel non-cAMP-dependent mechanisms elucidated from the multi-omics data play important roles in driving the C. albicans yeast-filament transition.

Reviewer #2: Page 6, the first paragraph. Using percentages alone to describe the ability of hyphal growth of the PR mutants can be misleading because it cannot accurately reflect the quality of hyphal growth. As shown in Figure 1B, most of the PR mutants show abnormal hyphal morphology. They are significantly shorter and thicker than WT hyphae. When you say 93% of PR18 cells form hyphae, it is not what the image shows. It is acceptable to say 93% of the cells respond to hyphal induction. The authors should modify the description.

Reviewer #3: 1. One question concerned the emergence of mutants and their placement in different classes. The authors classify the cyr1 revertants into 4 classes (1, 2, 3 and 3+). It was not clear how many PR revertants were analyzed and which ones fall into each class. Also, were truly independent revertants identified for each class? For example, it was striking that the two class 2 revertants appear to have the same genomic change. And is classification based on phenotypes and genotypes, or just phenotypes? Also, there was only one PR strain for classes 3 and 3+ - were more identified in these classes? Overall, there was confusion about identification and classification of PRs. (And was PR2 in this paper the same strain as PR2 in Parrino et al, 2017?)

2. In Fig. 1 and Fig. 2 the authors show nicely that PR strains form filaments in response to several cues. However, the quantification in Fig. 2D and E only shows the % filamentous in the presence of GlcNac, not in the control cultures (without GlcNac). It would be more convincing to see the % change in filaments +/- GlcNAc, to know how much more filamentation occurs with GlcNac relative to the control (similar to what the authors showed in the Parrino et al paper, Fig. 8B). Some strains appear to show a high % of background filamentation even without GlcNac, which gets to the heart of the question of how much filamentation occurs even without a stimulus in these strains.

3. In their previous paper on this topic (Parrino et al.), the authors said that no mutations involving BCY1 were identified in faster growing cyr1 pseudorevertants. Were mutations missed? Or were the monosomic regions (changes in copy number) not detected? While not a critical factor, clarification of this would be helpful.

4. Could the authors clarify how the bcy1/BCY cyr1/cyr1 strain was made? Was this using the CRISPR system or a different system? In general, more information about strain construction is required. This should include which oligos are used for which constructs / strains, as this is not obvious just from the oligos listed in Table S5. This information could be included in supplementary information if too extensive for the main methods. In addition, how were large CRISPR-mediated deletions confirmed? By PCR of junctions or by whole genome sequencing?

5. How does a bcy1/bcy1 cyr1/cyr1 double mutant behave? Has this been tried either by these authors or in other studies (either in Candida or other species)? It might also have been useful to add BCY1 back to the PR mutants to further establish the contribution of gene dosage to growth rates/filamentation, although it is acknowledged that slow growing transformants are likely to have to be identified.

6. One concern was how the phosphoproteomics data was being used/interpreted. Much of the paper’s more general conclusions rely on the ability to accurately identify all phosphosites and, in several cases, the kinases that produce these phosphosites. I am not convinced that a proteome-wide screen will detect all phosphosites (for technical reasons) and so it still seems possible that PKA activity is the same before/after hyphal induction? Along the same lines, it should be clarified that Yck2 and Hhc1 are predicted to phosphorylate some of the detected sites based on their consensus motifs, and that direct phosphorylation by these factors is not shown in the current study. (And attempts to make phospho-mutations in key residues in Bni1/Mob2 did not show hyphal defects).

7. For Table S3, information lacking as to what is in the table. Terms like “positions in master proteins” are not defined or what the code in this column refers to. Also, what are the units in columns J-Q for abundance?

8. The authors state that the hgc1 null mutant caused a stronger hyphal defect in the PR13 mutant than in the WT control, but this is hard to see from the cell images in Fig. S5A. Could a quantitative analysis be included? The same is true for the effect of yck2 deletion on WT/PR13 strain backgrounds. In the case of the yck2 deletion the strains look very pseudophyphal, so are authors distinguishing between hyphae and pseudohyphae, or are both just counted as filaments throughout? This is an important point that should be discussed as it is relevant to all of the filamentation assays.

9. The text says that translation rates of hyphal TFs increased dramatically during hyphal growth and cite Fig. 5E. However, the data in this panel is not significant for hyphal TFs as a group? This should be revised/clarified.

10. An overall model showing where things stand in terms of hyphal regulation would help and would be more useful than Fig. S6 which did not add much to the paper. This model could clarify what is truly new about the hyphal induction model, given that it seems like pseudorevertants are still dependent on Efg1 for hyphal formation (Parrino et al.) and that Hgc1 and Yck2 were previously shown to regulate filamentation. Is it true that all pseudorevertants still require Efg1 to filament like PR2 shown previously? Overall, the reader is left unclear as to how transcription factors and kinases are envisaged to work together, even if this is still hypothetical.

**Part III – Minor Issues: Editorial and Data Presentation Modifications**

Reviewer #1: 1. Most deletions by the CRISPR-Cas9 system are heterozygous. Can the authors confirm that all the chromosome deletions shown in Figure 2 are actually heterozygous mutations?

2. Lines 212-213. There is a discrepancy about Efg1 phosphorylation by the PKA pathway. Aside from pointing out this discrepancy it would be useful to have some explanation.

3. S3 Figure: why is the spt5 mutant reduced for filamentation but the srb9 spt5 mutant shows enhanced filamentation? Some additional commentary and explanation here would help.

4. S5 Figure: the phospho-mutants of Mob2 and Bni1 fail to show a filamentation defect. It would help to have some additional evidence that non-cAMP-dependent phosphorylation of target proteins is important for driving filamentous growth.

5. S3 Figure: the yck1 mutant in the WT background generates filaments but in the PR13 background this is not the case. Why? This should be discussed.

6. Figure S2A: Why was total protein used for normalization rather than a standard loading control (eg: Act1)? How could quantitation be performed if this not a single band on the gel?

Reviewer #2: Figure 2C. The wide variation in doubling times of the cyr1 mutant in six independent experiments is puzzling. Were different strains or colonies used? Or there are other explanations.

Lines 66-67 and 276-278. The authors argue that because the cyr1 deletion mutant grows very slowly, it does not support the essential role of Cyr1 in hyphal growth. The statements are not entirely correct. There are point mutations in the LRR domain of Cyr1 that allow normal yeast growth but not hyphal growth. Xu et al., Cell Host & Microbe 4:28-39; Wang and Xu, Commun Integr Biol 1, 137-139.

Line 76-77. The statement ‘transcriptional analysis of hyphal induced genes has not revealed the mechanism of hyphal induction’ is not fair. While it is true that the vast majority of HSGs are not required for hyphal growth, some of them play critical roles in hyphal development, such as UME6 and HGC1. This study of the PR mutants also confirms the essential role of HGC1 for hyphal development.

Line 86. It is not appropriate to use these studies to argue against the role of cAMP in regulating hyphal growth. Cdc28 is far downstream along the hyphal induction pathway; and the Cdc28/Hgc1 is activated by cAMP signalling. Please modify this statement.

Line 179. The authors should be careful when they say ‘cAMP signaling is not necessary for hyphal growth or transcription regulation of genes during hyphal induction’. cAMP signaling includes necessarily the targets of cAMP such as PKA. In the PR mutants, PKA is upregulated due to the partial loss of Bcy1, which could, in a way, support the importance of cAMP signalling (even only to ensure a basal level of activity). In a signal pathway, activation of a downstream component often renders upstream elements unnecessary, but it does not mean the pathway is not needed. Here, the authors merely demonstrate that the PR mutants retain the ability to activate hyphal-specific genes in response to their hyphal induction condition.

Lines 186, 190. The transcriptome profile similarities are relative. By eyeing the heatmaps, one sees more differences than similarities even between the most similar ones. The analyses merely show that some are more similar than others. The description ‘similar patterns’ or “highly similar’ is arbitrary and should be rephrased.

Lines 303-305. The constitutive growth of the TUP1 mutant does not argue against the role of TFs in hyphal growth. In the absence of the general repressor Tup1, some of the hyphal-promoting TFs are upregulated.

Figure 3c legend. The ‘Grey boxes’ are not obvious. I only see different shades of blue and yellow. It is better to state that values above or below a specific number are insignificant. The results (the CYR1 row) show that CYR1 mRNA was detected in the CYR1-deletion and PR mutants, albeit at very low levels?

Figure 4c is incomplete. Readers want to see the comparison between WT+GlcNac vs. WT-control and PR13+GlcNac vs. PR13-control for substrates of both Cdc28 and Yck2 kinases.

Was only the WT strain used in the experiments described in Figure 5? Are there any evidence that the protein level of these TFs are increased in the PR mutants?

Reviewer #3: 1. I suggest the authors could consider renaming the strains so that they are more informative as random numbers (PR13, PR12, PR18 are not meaningful). The new names could match the class from which they are from 1-1, 2-2 etc….

2. Lines 212-213. Are the authors saying “inconsistent” or “consistent” ?

3. Wrong figure is cited on line 169 - should be Fig S2.

4. The gene name for Yck2 on line 232 is confusing as says “yeast casein kinase 1 (Yck2)”. This could be clarified by saying casein kinase I family?

5. In many figures the statistical significance is shown relative to the WT or parental strain, but it is also relevant to see significance relative to other strains in a series. For example, in Figure 2 it would be interesting to see the pseudorevertants compared to the cyr1 mutant as well as the WT. This was also true of other panels in the paper.

6. In the Author Summary and Discussion, it is stated that translational regulation of key transcription factors is a novel mechanism of hyphal regulation. However, translational regulation of several of the key TFs has previously been shown so that is not a truly novel finding. A revision of the text would help here.

7. On 237, the text reads that “… both kinases [Cdc28 and Yck2] phosphorylated 11 sites in 8 genes….. Kinases phosphorylate proteins.

8. Could the authors clarify why the strains were first grown in SC+ Galactose media for transcriptomic analysis and also for proteomics analysis?

9. In discussing the SRB9 and SPT5 data it would be relevant to include a mention that SSN3 mutants also impact filamentation phenotypes, as there seems to be a pattern emerging with mutation of transcriptional co-factors impacting filamentation (Wartenburg et al., 2014).

PLOS authors have the option to publish the peer review history of their article (what does this mean?). If published, this will include your full peer review and any attached files.

Reviewer #1: No

Reviewer #2: **Yes: **Yue Wang

Reviewer #3: No
---

## [Decision Letter · Decision Letter 1]

5 Jul 2021

Dear Dr. Konopka,

Thank you very much for submitting your revised manuscript "Integrative multi-omics profiling reveals cAMP-independent mechanisms regulating hyphal morphogenesis in Candida albicans" for consideration at PLOS Pathogens. The manuscript was reviewed by members of the editorial board and by two of the previous reviewers. The reviewers did not suggest additional experimentation, but noted a number of matters that require rewriting and inclusion of additional information to improve clarity. Based on the reviews, we are likely to accept this manuscript for publication, providing that you modify the manuscript according to the review recommendations.<o:p></o:p>

<o:p> </o:p>

Please pay especially careful attention to the following. The present manuscript seems to contradict the conclusion of your previous paper (ref. 14) that mutations outside the Cyr1-cAMP-PKA pathway can suppress the cyr1 mutant phenotype. In contrast, all analyzed mutants of the new study are classical suppressor mutants in a downstream component of the Cyr1-cAMP pathway, including one of the previously described mutants (PR2). Therefore, the findings of the present work actually confirm the established model that the role of Cyr1 is to allow PKA activity. As already pointed out in our decision letter to the original manuscript, it is important to make this clear to the reader. The information that one of the previously described mutants in fact has lost one of the BCY1 alleles should not be hidden in the methods section (see also comment by reviewer 3), but highlighted in the results section. Furthermore, we again request that you make a clear statement as to whether, upon re-analysis, the other mutants PR1 and PR3 described in ref. 14 also contain mutations affecting Bcy1 or if these indeed contain suppressor mutations that bypass the requirement for Cyr1 in a way that does not restore PKA activity.<o:p></o:p>

<o:p> </o:p>

We also note that, in the light of the new data showing the high filamentation rates of the mutants under non-inducing conditions, the statement in lines 169-173 is not correct any more. The 180-270 kb deletion roughly doubled the filamentation rate from ca. 30% to ca. 60% under non-inducing conditions, which appears to be a stronger effect than the minor increase seen upon addition deletion of the 10-kb region from 250-260, which led to the conclusion that genes in this region are involved in the regulation of morphogenesis. This should be corrected and critically discussed, considering that all mutants were generated only once and minor differences might not be reproduced with independently generated strains.<o:p></o:p>

<o:p> </o:p>

Please address these and the reviewers’ other comments, as there will be no further round of revision. We look forward to receiving an appropriately modified manuscript that is suitable for publication in PLoS Pathogens.<o:p></o:p>

Sincerely,

Joachim Morschhäuser

Associate Editor

PLOS Pathogens

Alex Andrianopoulos

Section Editor

PLOS Pathogens

Kasturi Haldar

Editor-in-Chief

PLOS Pathogens

orcid.org/0000-0001-5065-158X

Michael Malim

Editor-in-Chief

PLOS Pathogens

orcid.org/0000-0002-7699-2064

Reviewer Comments (if any, and for reference):

Reviewer's Responses to Questions

**Part I - Summary**

Reviewer #1: In general, the authors have done a good job in addressing my concerns and the revised manuscript is significantly strengthened. However, the authors have still not provided independent validation for the multi-omics data as requested. While it is reassuring to know that these data are highly consistent, the exact number of replicates was not clearly specified. From the legend for Figure 4B it appears that only 2 replicates were used for the proteomics and phosphoproteomics experiments.

Reviewer #3: The revised version of the Min et al. manuscript addresses several of the issues raised by reviewers in the original submission, and the edited version is much improved. There were still some points that could use further clarification as outlined below. Most of these request editing of the text or data analysis rather than additional experiments.

**Part II – Major Issues: Key Experiments Required for Acceptance**

Reviewer #1: (No Response)

Reviewer #3: 1. The authors emphasize that the number of phosphopeptides assigned to PKA activity do not change between yeast and hyphae. But are the exact same phosphopeptides present in both conditions? It appears that GSEA analysis shows that the functional class of proteins phosphorylated do not change, but is it possible a change in the PKA pattern could be important even if the total number of PKA phosphopeptides do not? This was hard to tell by looking at the tables.

2. The recent Mundodi et al paper (PMC7849906) that examines translation efficiencies associated with filamentation should be mentioned/discussed. Some comparison of the current data set with the Mundodi one would be interesting as to whether the same genes/trends are identified. A cursory comparison appears to show different trends, albeit using different experimental approaches. While the current paper shows that the protein-to-mRNA ratio of UME6, SFL1, BRG1, TEC1, and FKH2 increased dramatically during hyphal induction (while mRNA levels did not), Mundodi shows that the TE (translational efficiency) for several of these genes decreased during hyphal induction (e.g. TEs for UME6, SFL1, BRG1, FKH2) although the decrease was not significant. Can the authors explain why TE is reduced or unchanged, whereas protein/mRNA ratios increase for these genes? Is this due to induction of hyphae using different conditions? This makes it hard to know whether translation of these regulatory genes is really up or down under hyphal conditions.

3. In terms of the overall model, could the authors clarify what is meant by a new “physiological state” that supports hyphal formation? Presumably transcription of certain targets is necessary to prime the cell, but this priming is seen as necessary but not sufficient for hyphal induction in this model? Otherwise this term seems a but mysterious.

**Part III – Minor Issues: Editorial and Data Presentation Modifications**

Reviewer #1: (No Response)

Reviewer #3: One remaining question still concerns how cyr1 revertants were divided into the 4 PR classes. It is still unclear if these are being classified based off growth rates and filamentation percentages alone, or if other phenotypes are being use for classification (or if genotypes are also used). This needs to be described clearly for the reader. It is also still unclear if multiple PR revertants were identified for each class of the classes or if some classes are really singletons.

The authors state that on lines 50-52 that “post-transcriptional mechanisms regulate the levels of a set of key transcription factors that are important for hyphal induction, suggesting a novel form of translational regulation.” I am still not sure that this can be considered “novel” given that post-transcriptional mechanisms have already been linked to some of these genes (via their UTRs) and that the mechanism involved for other transcription factors has not been identified. I would simply suggest that removing the term “novel” would be safer to avoid the risk of seeming to over sell the conclusions.

The authors provide data showing that some of the PR strains are highly filamentous even without an inducing cue and it seems more could be made of this point. It appears important that basal rates of filamentation are now very high in some PR classes such as in class 2/PR12.The clarity of the phosphorylation proteomics data is improved over the initial submission (Data S3). However, it was not clear to me why not all rows have a peptide (15mer) listed in column X? Or why some rows have multiple 15 mers listed? Also in Data S3 it shows that some peptides are phosphorylated at multiple positions – sometimes these positions are indicated in column D and sometimes they are not?

Data table S4 needs a similar ReadMe introduction to Data table S3 to provide clarity as to what is being shown.

It would be useful if the methods included the volumes of cultures/cells used for transcriptomic and proteomic analyses, as well as for western blotting analyses.

Regarding the PR2 mutant that was previously described as not have any genetic changes linked to phenotypic differences, it does appear the authors hide this fact by only adding a sentence to the Methods (463-465). This really should be mentioned more plainly in the Results and not in the Methods where most readers would not find this fact. I do not see this as a problem that needs to be hidden.

PLOS authors have the option to publish the peer review history of their article (what does this mean?). If published, this will include your full peer review and any attached files.

Reviewer #1: No

Reviewer #3: No

Figure Files:

Data Requirements:

Reproducibility:

References:

---

## [Editor Report · Decision Letter 2]

2 Aug 2021

Dear Dr. Konopka,

We are pleased to inform you that your manuscript 'Integrative multi-omics profiling reveals cAMP-independent mechanisms regulating hyphal morphogenesis in Candida albicans' has been provisionally accepted for publication in PLOS Pathogens.

Best regards,

Joachim Morschhäuser

Associate Editor

PLOS Pathogens

Alex Andrianopoulos

Section Editor

PLOS Pathogens

Kasturi Haldar

Editor-in-Chief

PLOS Pathogens

orcid.org/0000-0001-5065-158X

Michael Malim

Editor-in-Chief

PLOS Pathogens

orcid.org/0000-0002-7699-2064
---

## [Editor Report · Acceptance letter]

11 Aug 2021

Dear Dr. Konopka,

We are delighted to inform you that your manuscript, "Integrative multi-omics profiling reveals cAMP-independent mechanisms regulating hyphal morphogenesis in Candida albicans," has been formally accepted for publication in PLOS Pathogens.

Best regards,

Kasturi Haldar

Editor-in-Chief

PLOS Pathogens

orcid.org/0000-0001-5065-158X

Michael Malim

Editor-in-Chief

PLOS Pathogens

orcid.org/0000-0002-7699-2064